# GPEN: Global Positional Encoding Network for Graphs

## Abstract

Non-grid-structured data, *e.g.*, citation networks, social networks, and web page networks, is often represented as graphs. However, such data cannot fit into Convolutional Neural Networks (CNNs) like images because of the variable size of unordered nodes and the uncertain number of neighbours for each node. Thus, Graph Neural Networks (GNNs) have been designed. They use a message-passing scheme to aggregate each node's and its neighbours' feature representations, regardless of the number of nodes and their order. Introducing feature-independent encoding methods to GNNs is crucial to preserving graphs' structural information and making node representations more discriminative. However, local-distance-aware methods, *e.g.*, DE-GNN, only contain the information within subgraphs, resulting in ambiguity when comparing two subgraphs with the same structure. In this paper, our Global Positional Encoding Network (GPEN) is proposed to embed each node's global positional information by calculating their distances to a set of randomly sampled referential nodes. We employ contrastive loss on pairwise distances of different nodes to make positional representations more discriminative while retaining the relative interactions between nodes. We assessed our GPEN on node classification tasks by incorporating the encoding method into backbone GNNs. Our results demonstrate that it exceeds state-of-the-art encoding methods on GNN benchmark datasets with up to 34.26% accuracy.

## 1 Introduction

Graph Neural Networks (GNNs) have become famous for analyzing non-grid-structured data, like citation networks, social networks, and web page networks, which can be represented as graphs Kipf & Welling (2017); Yanardag & Vishwanathan (2015); Pei et al. (2020); Xu et al. (2019); Veličković et al. (2018); Hamilton et al. (2017). GNNs employ a message-passing scheme to recursively aggregate a central node's and neighbours' features in graphs. Specifically, node/edge representations are passed through the edges to their nearby nodes. Then, these representations are aggregated as new central node representations using permutation-invariant functions and learnable parameters. This process is repeated $k$ times, resulting in a feature vector representing the central node, capturing the structural information and node feature distribution within the central node's $k$-hop neighbourhood Hamilton et al. (2017).

Although the message-passing scheme helps GNNs gather structural information based on node and edge attributes, and enables GNNs to perform well in specific domains, that ability becomes limited when applied to more diverse applications, such as forecasting passenger flow levels in airports, predicting airline connections, or classifying social networks that lack node and edge attributes Yanardag & Vishwanathan (2015); Ribeiro et al. (2017); Zhang & Chen (2018). Some strategies have been proposed to address this limitation. They rely on attribute-independent and deterministic features, such as hand-crafted rules Zhang & Chen (2018); You et al. (2021) and random-walk-based probability Li et al. (2020), to encode the distance between central nodes and their neighbours in a given locality. The goal is to learn the topological nature of graphs or subgraphs as a unique representation based on the encoded distance, which can then be used for various downstream tasks. Encoding the distance of a central-neighbour pair as a vector and combining it with the neighbour's attributes during message-passing can effectively learn local structural information. This technique has been proven to enhance the accuracy of GNNs in node classification tasks Yin et al. (2020). However, we argue that local-distance-aware encoding approaches fail to deliver discriminative features when

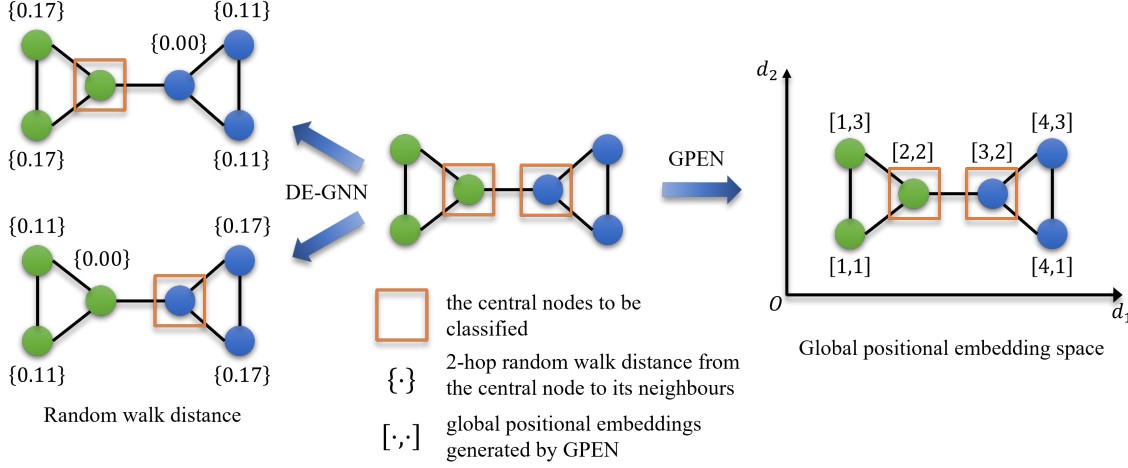

Figure 1: Intuitive comparison of DE-GNN and GPEN. **Central**: A symmetric graph where the colour of the nodes indicates their classes. **Left**: Given one central node, DE-GNN uses 2-hop random walk distance from the central node to its neighbours as node features. In this case, the encoding method generates identical features for the central nodes. **Right**: To distinguish the central nodes, GPEN generates unique global positional embeddings through contrastive learning.

two subgraphs have a similar structure but are located in different parts of the same graph, as the left part of Figure 1 shows. As a result, these non-discriminative features will harm the expressiveness of GNNs when the ground truth labels of those nodes are different. Therefore, learning the global position of each node, which contains structural information and makes similar subgraphs possible to distinguish from each other, is more important.

In this paper, we present the Global Positional Encoding Network (GPEN), which generates a distinct global positional embedding for each node in a graph based on graph structure without needing either node or edge attributes, as demonstrated in Figure 1. Unlike DE-GNN Li et al. (2020) employing relative distances between a central node and its neighbours, GPEN randomly chooses nodes from the graph as a referential node set. Each node's global position in the graph is determined by the set of vectors representing random walk probabilities between the node and the referential nodes. For each node, GPEN embeds the probability-like vectors as a positional embedding, which allows the downstream networks to learn the structure representations. We utilize self-supervised learning He et al. (2020) to ensure that the positional embeddings are consistent and able to distinguish between different nodes. We apply a contrastive loss to learn the embeddings. In this approach, we consider embeddings of the same node under different reference sets as positive pairs and aim to maximize their similarity.

In a nutshell, our contributions are summarized as follows:

1. We reveal that distance encoding approaches have limitations in providing useful graph structure information for the node classification task on homophilic graphs.

2. We propose a framework named GPEN to generate a global positional embedding for each node according to a referential node set, without needing either node or edge attributes.

3. We introduce contrastive learning to the progress of training the GPEN, which increases the stability of the global positional embedding.

We use classic GraphSAGE and transformer-based NAGphormer as the backbone GNNs and combine them with GPEN and various positional or distance encoding methods. Our experiments on the node classification task have shown that GPEN significantly improves the results for datasets facing the challenges mentioned earlier. GPEN generates global positional embeddings that can effectively represent the structural information of a graph, even when node attributes are absent.

## 2    Related Work

**GNNs.** Given a central node in a graph in each layer of a GNN, the message-passing scheme generates a new node feature by aggregating features from neighbours of the central node and mapping them to a new feature space. Following this principle, many GNN designs are proposed and applied to various tasks. GCN maps features by approximating a graph convolution filter Kipf & Welling (2017). GAT applies the self-attention mechanism to the aggregation function Veličković et al. (2018). GraphSAGE concatenates central node features and aggregates neighbour features before passing them to the mapping function Hamilton et al. (2017). Regardless of the variety of aggregation and mapping functions, their corresponding computational graphs determine the final node representations containing structural information, as shown in Figure 2. Thus, the expressiveness of the GNN is limited if the computational graphs of nodes are not distinctive enough. This limitation becomes even more significant when node/edge attributes are unavailable as input.

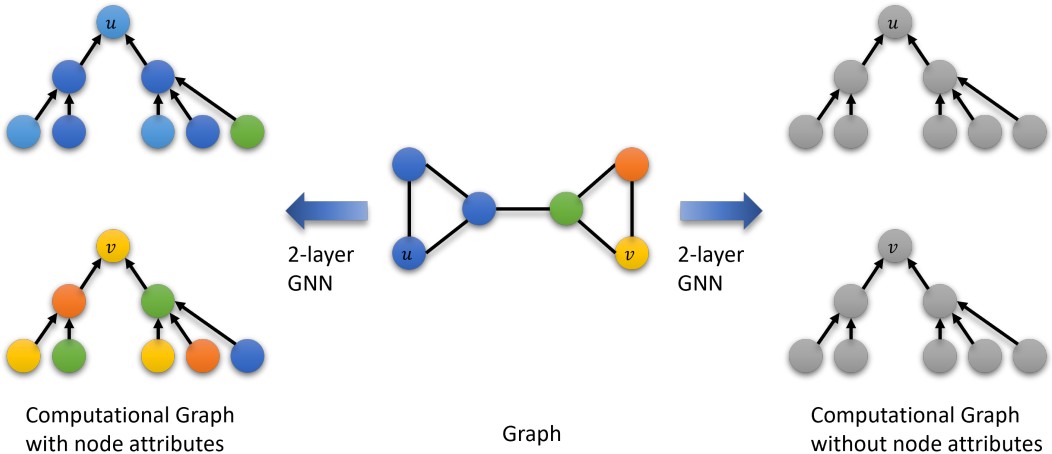

Figure 2: When passing a graph to a GNN, the computational graph shows the aggregation path of neighbours' features to the central node. **Left**: The computational graphs of the node $u$ and $v$ are distinctive when having node attributes. **Right**: The computational graphs of the node $u$ and $v$ are the same when no node attributes are available.

**Distance and Positional Encoding.** Introducing features independent of node/edge attributes to GNNs is a way to enhance their ability to distinguish between different instances. Two types of representations that meet this requirement are distance encoding (DE) and positional encoding (PE). Usually, GNNs learn local or global structure representations from aggregating these embeddings and then use them in node/edge/graph level tasks.

DE approaches aim to define the distances between the central node and its $k$-hop neighbours in subgraphs. For example, to predict the existence of a link between two target nodes, SEAL labels other nodes surrounding target nodes according to hand-crafted rules based on the shortest path distance Zhang & Chen (2018). DE-GNN encodes the distance between each node and a target node set in a subgraph using random walk probabilities Li et al. (2020). The target node set is the target of the task, such as single nodes (node classification), node pairs (link prediction), and node sets (triangle prediction). Although using positional encoding like Laplacian eigenmaps Belkin & Niyogi (2003) to generate the positional feature for each node, PEG maps the features between the end nodes to their edge as weights, which act as distances Wang et al. (2022). Such mapping can reduce global structure information inherent in Laplacian eigenmaps and produce similar weights at the symmetric positions in a graph. So, we classify PEG as a distance encoding method. These local-distance-aware methods have a common drawback: structure representations are indistinguishable in the case of similar subgraphs.

On the other hand, PE approaches lean toward a unique global positional embedding for each node, which implicitly contains structural information while making the representation diverse. Dwivedi et al. (2020) assigns a Laplacian eigenvector to each node as their positional encoding, but this method suffers from

instability of multiple eigenvalues, and the computational complexity limits the application on large-scale graphs. ID-GNN colours the ego network's central node, indicating its position You et al. (2021). P-GNN encodes each node's position in the graph into a low dimensional metric space through random walk probabilities to a randomly selected node set You et al. (2019). Following Bourgain's Theorem Bourgain (1985), this method provides a less computationally complex and relatively stable way to generate global positional representations. However, the length of P-GNN's position embeddings depends on the number of nodes, and it only uses the embeddings on edge-level tasks, which limits its capability as a general solution that can be used as a pre-trained model.

**Contrastive Learning.** Contrastive Learning aims to capture the invariant representation of different data augmentation He et al. (2020); Chopra et al. (2005); Chen et al. (2020); Chen & He (2021); Zhang et al. (2022), *e.g.*, low-level data augmentation of an image, while keeping the semantic meaning. Technically speaking, it pulls the positive samples closer while pushing negative samples away to achieve such invariance through optimizing the InfoNCE loss Oord et al. (2018). Since contrastive learning is independent of downstream tasks, it can be naturally applied to graph-based tasks. GCC Qiu et al. (2020) samples ego networks based on random walks to augment the graph structures for contrastive learning. MVGRL Hassani & Khasahmadi (2020) uses graph diffusion and subgraph sampling to generate different views of the same graph. GraphCL You et al. (2020) develops contrastive learning for GNN pre-training and proposes augmenting the graph through edge dropping and perturbation Rong et al. (2019). Those methods adopt the InfoNCE to find invariant feature representations via perturbing the original graph structure with different augmentation methods, failing to find a unique position representation for each node in different graph structures.

## 3 Method

Given a graph $\mathcal{G} = (\mathbb{V}, \mathbb{E})$, where $\mathbb{V}$ is a node set and $\mathbb{E}$ is an edge set, we define a global positional embedding space. In this space, all nodes are assigned a unique position that reflects their global positions in the graph. We propose GPEN to learn this global positional embedding space from pre-defined pair-wise node distances in the graph, which provides a more powerful encoding for describing structural information. As Figure 3 shown, GPEN generates a global position embedding for each node based on its pair-wise node distances to a randomly selected node set, which can be concatenated with the node attribute and sent to any GNN model for arbitrary tasks. Additionally, we apply a contrastive learning loss to the embeddings to ensure they are unique and consistent with each node's global position under different node set selections.

### 3.1 Relative Distance in Graph

Firstly, we define the distance between any two nodes in a graph to describe the local structure. The distance should be permutation-invariant because the same graph with different orders of the nodes' indices can result in different adjacency matrices.

**Definition 3.1** *Given any two nodes $u, v \in \mathbb{V}$ in the graph $\mathcal{G}$, the k-step random walk probability from $u$ to $v$ is defined as:*

$$p^{(k)}(v|u) = (\boldsymbol{W}^k)_{uv}, \tag{1}$$

$$(\boldsymbol{W})_{uv} = \begin{cases} (\boldsymbol{A})_{uv} \big( \sum_{v=1}^{|\mathbb{V}|} (\boldsymbol{A})_{uv} \big)^{-1}, & \text{if } uv \in \mathbb{E}, \\ 0, & \text{otherwise,} \end{cases} \tag{2}$$

*where $\boldsymbol{A}$ is the adjacency matrix of $\mathcal{G}$.*

We add multiple random walk probabilities from node $u$ to $v$ with different steps to obtain more local structure information. As $\boldsymbol{W}$ can be asymmetric, we choose a vector $\boldsymbol{y}_{uv} \in \mathbb{R}^{2k}$ to describe the relative distance from node $u$ to $v$ in graph $\mathcal{G}$. $\boldsymbol{y}_{uv}$ consists of a sequence of k-step random walk probabilities from node $u$ to $v$ and $v$ to $u$ as:

$$\boldsymbol{y}_{uv} = \left[ p^{(1)}(v|u), ..., p^{(k)}(v|u), p^{(1)}(u|v), ..., p^{(k)}(u|v) \right]. \tag{3}$$

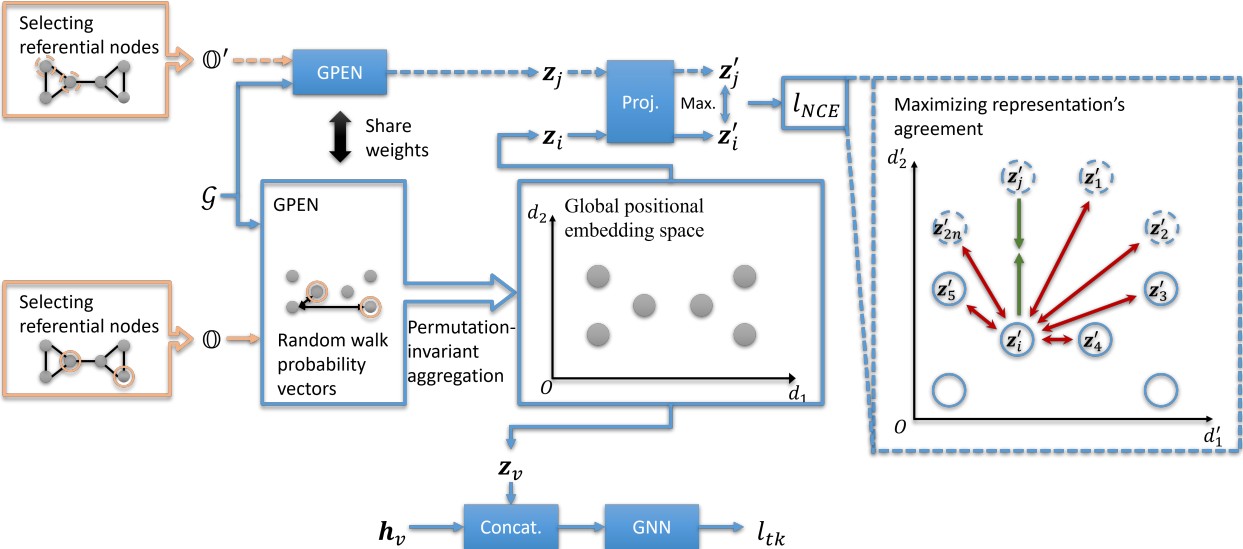

Figure 3: The general scheme of our Global Positional Encoding Network (GPEN). Given a graph $\mathcal{G} = (\mathbb{V}, \mathbb{E})$, a referential node set $\mathbb{O} \in 2^{\mathbb{V}} \setminus \varnothing$, and a node $v$, GPEN uses random walk probability vectors to describe the relative distances between $v$ and each node in the $\mathbb{O}$. Then, GPEN generates a global positional embedding $\boldsymbol{z}_v$ using a permutation-invariant aggregation function with learnable parameters. $\boldsymbol{z}_v$ can either be concatenated with the node attribute of $v$, $\boldsymbol{h}_v$, or as the only input feature and sent to the backbone GNN. To keep the consistency of global positional embeddings of the same node under different selections of referential node set, GPEN generates $\boldsymbol{z}_i$ and $\boldsymbol{z}_j$ for the same node from two different referential node sets $\mathbb{O}$ and $\mathbb{O}'$, respectively. Then, we map the embeddings to a representation space as $\boldsymbol{z}_i'$ and $\boldsymbol{z}_j'$ by a nonlinear projection head and employ InfoNCE loss, aiming to maximise these representations' agreement. In this space, representations of the same node under the different referential node selections are pulled towards each other. The representations of different nodes are pushed away.

## 3.2 Global Positional Encoding Network

However, there is still a limitation that the relative distance vector $\boldsymbol{y}_{uv}$ is insufficient to provide structural information to distinguish any two target nodes. We first define a simple permutation-invariant function $f : \mathbb{Y}_v^{sub} \to \mathbb{R}^n$ that extracts the structural information of a subgraph $\mathcal{G}_{sub} = (\mathbb{V}_{sub}, \mathbb{E}_{sub})$ into an $n$-dimensional vector and assigns it to a target node $v$:

$$h(\mathbb{Y}_v^{sub}) = \sum_{\boldsymbol{y}_{uv} \in \mathbb{Y}_v^{sub}} f(\boldsymbol{y}_{uv}), \tag{4}$$

where $\mathcal{G}_{sub}$ induced by $n_{hop}$-hop neighborhood of the target node $v$, $\mathbb{V}_{sub} \subseteq \mathbb{V}$, $\mathbb{E}_{sub} \subseteq \mathbb{E}$. $\mathbb{Y}_v^{sub} = \{\boldsymbol{y}_{uv} | u \in \mathbb{V}_{sub}\}$ is a multiset. Given two target nodes $v_1$ and $v_2$, a GNN cannot distinguish them by using their structural information $h(\mathbb{Y}_{v_1}^{sub})$ and $h(\mathbb{Y}_{v_2}^{sub})$ alone when $\mathbb{Y}_{v_1}^{sub} = \mathbb{Y}_{v_2}^{sub}$. This limitation will harm the representation ability when labels of $v_1$ and $v_2$ differ. Thus, we design the GPEN inspired by P-GNN You et al. (2019) to address this limitation.

**Definition 3.2** *Selecting a set of nodes $\mathbb{O} = \{o_1, o_2, ..., o_{N_r}\} \in 2^{\mathbb{V}} \setminus \varnothing$, $N_r = |\mathbb{O}|$, the global position of a node $v \in \mathbb{V}$ in the graph $\mathcal{G}$ can be defined as a multiset of relative distance vectors: $\mathbb{Y}_v = \{\boldsymbol{y}_{ov} | o \in \mathbb{O}\}$. Nodes composing the set $\mathbb{O}$ are named referential nodes.*

When $\mathbb{O}$ is appropriately selected, $\mathbb{Y}_{v_1}$ can be different to $\mathbb{Y}_{v_2}$ even if $h(\mathbb{Y}_{v_1}^{sub}) = h(\mathbb{Y}_{v_2}^{sub})$. The intuition of selecting referential nodes is that each of them is sampled from the whole graph instead of a subgraph. It serves as a coordinate axis that breaks the similarity of the local structure and can be used to encode a unique and global position for all nodes in the graph.

To encode the global position, we propose the Global Positional Encoding Network (GPEN), which generates a global positional embedding for each node:

$$\boldsymbol{z}_v = f_{GPEN}(\mathbb{Y}_v) = f_2\Big( \sum_{\boldsymbol{y}_{ov} \in \mathbb{Y}_v} f_1\big(\boldsymbol{y}\big) \Big), \tag{5}$$

where $f_1$ and $f_2$ are two two-layer MLPs with a non-linearity. $\boldsymbol{z}_v \in \mathbb{R}^{d_{GP}}$. $d_{GP}$ is the dimension of the global positional embedding. Then, global positional embeddings can be individually used or concatenated with node attributes as new inputs of any GNNs. In this paper, we uniformly select $N_r$ nodes at random in a graph as referential nodes.

### 3.3 Contrastive Loss

Our goal is to enable GPEN to learn unique global positional embeddings for all nodes in a graph. Using these positional embeddings, the following GNN can generate discriminative local structure representations of subgraphs in the same graph, even if they have the same structure. Meanwhile, the referential node set $\mathbb{O}$ defines the global position, which is randomly selected. Thus, maintaining consistency in the global positional embedding of the same node is essential. To achieve these two goals, P-GNN follows Bourgain's Theorem and samples almost all nodes in a graph multiple times. However, P-GNN loses the flexibility of applying the embedding to arbitrary tasks while minimizing the distortion of the positional embedding. In contrast, we only select a small number of unrepeated nodes and employ the InfoNCE loss Oord et al. (2018) to GPEN during the training stage to meet these two goals.

Inspired by SimCLR Chen et al. (2020), for each target node in the same minibatch with $N$ samples, a projection head $f_{Proj} : \boldsymbol{z}_v \to \mathbb{R}^{d_{Proj}}$ first maps the global positional embedding to a $d_{Proj}$-dimensional representation space:

$$\boldsymbol{z}'_v = f_{Proj}(\boldsymbol{z}_v). \tag{6}$$

We choose a two-layer MLP with nonlinearity as the projection head in this paper. As Figure 3 shows, we separately pass the same batch of nodes with two different referential node sets, $\mathbb{O}$ and $\mathbb{O}'$, to GPEN, then acquiring projections $\boldsymbol{z}'_{\mathbb{O}}$ and $\boldsymbol{z}'_{\mathbb{O}'}$ corresponding to $\mathbb{O}$ and $\mathbb{O}'$, resulting in a set $\{\boldsymbol{z}'_k | k = 1, 2, ..., 2N\}$. Given a pair of positive samples $\boldsymbol{z}'_i, \boldsymbol{z}'_j \in \{\boldsymbol{z}'_k\}$, which are from the same node, the purpose of the contrastive prediction task is identifying $\boldsymbol{z}'_j$ in $\{\boldsymbol{z}'_k\} \setminus \{\boldsymbol{z}'_i\}$ for $\boldsymbol{z}'_i$. Therefore, the InfoNCE loss is chosen to minimize the distance of the positive pair $(\boldsymbol{z}'_i, \boldsymbol{z}'_j)$ and push other negative samples away:

$$l_{NCE} = \frac{1}{2N} \sum_{i=1}^{2N} -\log \frac{\exp\big(d(\boldsymbol{z}'_i, \boldsymbol{z}'_j)/\tau\big)}{\sum_{k=1, k \neq i}^{2N} \exp\big(d(\boldsymbol{z}'_i, \boldsymbol{z}'_k)/\tau\big)}, \tag{7}$$

$$d(\boldsymbol{z}'_i, \boldsymbol{z}'_j) = \frac{\boldsymbol{z}'_i \boldsymbol{z}'^T_j}{\|\boldsymbol{z}'_i\| \|\boldsymbol{z}'_j\|}, \tag{8}$$

where $\tau$ is a temperature parameter.

In the end, the total loss $l_{tot}$ is formulated as:

$$l_{tot} = l_{tk} + l_{NCE}, \tag{9}$$

where $l_{tk}$ is the loss used in the original task, *e.g.*, cross-entropy loss.

## 4 Experiments

### 4.1 Datasets

In this paper, we evaluate our models on six public datasets for node classification tasks. All graphs are undirected and contain self-loops. The statistics of the datasets are listed in Table 1.

### 4.1.1 Citation networks

**Cora and Citeseer.** They are widely used citation networks proposed by Sen et al. (2008) where each node represents a paper, and edges denote the relationship of citations between papers. Node attributes are the bag-of-words representation of papers, and node labels are the academic topics of papers.

### 4.1.2 The new heterophilic datasets

We use the other four datasets modified or proposed by Platonov et al. (2023). These datasets are carefully formed and focus on heterophilic graphs with varying properties, which makes them better than problematic datasets such as Wikipedia network Rozemberczki et al. (2021).

**Roman-empire.** This dataset uses the most extended Roman Empire article from English Wikipedia, with the data from the 2022 Wikipedia dump Leskovec & Krevl (2014). Nodes in this graph represent individual words from the article, connected if they are sequential or part of the same syntactic structure. The syntactic roles of nods are acquired by spaCy Honnibal et al. (2020). The node features are represented by fastText word embeddings Grave et al. (2018). This dataset provides a chain network with low homophily, sparse connectivity, and potential long-range dependencies. The task is to classify the syntactic role of each node.

**Amazon-ratings.** The Amazon-ratings graph is based on data collected by Leskovec & Krevl (2014). It represents products as nodes, and edges are formed between products frequently co-purchased by the same customer. Node features are derived from the mean of fastText embeddings Grave et al. (2018) of product descriptions, and the task is to predict the product's average rating.

**Minesweeper.** Minesweeper is a synthetic dataset generated by Platonov et al. (2023). Nodes in the Minesweeper dataset represent cells in a Minesweeper game, edges connect neighbouring cells, and node features are represented by one-hot codes indicating the number of adjacent mines. There are 20% nodes randomly chosen as mines, but 50% nodes' features are unrevealed and assigned a separate binary feature. The classification task involves predicting whether a cell contains a mine.

**Tolokers.** This dataset models user interactions within the Toloka crowdsourcing platform Likhobaba et al. (2023), where nodes represent users and edges indicate tasks these users jointly performed in 13 projects. Node features include users' profile information and task performance statistics, and the task is to classify whether a user is banned in one of the projects.

To identify properties of datasets, we introduce the adjusted homophily and label informativeness proposed by Platonov et al. (2024). First, the edge homophily is defined by Zhu et al. (2020):

$$h_{edge} = \frac{|\{(u,v) \in \mathbb{E} : c_u = c_v\}|}{|\mathbb{E}|}, \tag{10}$$

where $c_u$ and $c_v$ are the labels of the node $u$ and $v$, respectively. However, the edge homophily is poorly measured when the classes are unbalanced. On the contrary, the adjusted homophily can be applied consistently across various datasets, regardless of differences in class numbers and size distributions. It is formulated as:

$$h_{adj} = \frac{h_{edge} - \sum_{i=1}^{C} D_i^2 / (2|\mathbb{E}|)^2}{1 - \sum_{i=1}^{C} D_i^2 / (2|\mathbb{E}|)^2}, \tag{11}$$

$$D_i := \sum_{v:c_v=i} d(v), \tag{12}$$

where $d(v)$ denotes the degree of the node $v$. A dataset with a larger $h_{adj}$ value tends to be homophilic, meaning nodes with the same label tend to connect to each other and form a cluster. In turn, a graph with a small $h_{adj}$ value is heterophilic.

To measure the degree of connectivity patterns existing in graphs, Platonov et al. (2024) introduce the label informativeness by using the normalized mutual information:

$$\text{LI} := I(c_\zeta, c_\eta) / H(c_\zeta), \tag{13}$$

given an edge $(\zeta, \eta) \in \mathbb{E}$ sampled uniformly at random among all edges. $c_\zeta$ and $c_\eta$ are random variables of labels of the node $\zeta$ and $\eta$, respectively. Thus, $LI \in [0, 1]$. $c_\zeta$ can be uniquely reconstructed from $c_\eta$ when $LI = 1$. $c_\zeta$ and $c_\eta$ are independent when $LI = 0$.

Table 1: Dataset statistics.

| Dataset | Cora | Cite. | Roma. | Amazon. | Mine. | Tolokers |
|---|---|---|---|---|---|---|
| # Nodes | 2708 | 3327 | 22662 | 24492 | 10000 | 11758 |
| # Edges | 5429 | 4732 | 32927 | 93050 | 39402 | 519000 |
| # Features | 1433 | 3703 | 300 | 300 | 7 | 10 |
| # Classes | 7 | 6 | 18 | 5 | 2 | 2 |
| $h_{adj}$ | 0.77 | 0.68 | -0.05 | 0.14 | 0.01 | 0.09 |
| LI | 0.59 | 0.46 | 0.11 | 0.04 | 0.00 | 0.01 |

## 4.2 Experimental Setup

To evaluate the effectiveness of our GPEN, we select various DE and PE approaches for comparison, including DE-GNN, PEG, and P-GNN. Following Yin et al. (2020), we use a two-layer GraphSAGE as a backbone model, with the same configuration for each dataset. We also use a recent transformer-based GNN named NAGphormer Chen et al. (2023) as another backbone model. NAGphormer uses Laplacian eigenvectors as PE and tokenizes $n_{tok}$-hop neighbours through a parameter-free aggregation, making it efficient for large graphs. Using NAGphormer, we can easily switch between different PE approaches and compare their performance. To ensure a fair comparison with the GraphSAGE-based results, we use a single layer of NAGphormer and set $n_{tok} = 2$ to ensure they have the same receptive field size of 2.

DE and PE embeddings from GPEN, DE-GNN, and P-GNN are concatenated with node attributes along the channel dimension. Particularly, the referential node set in GPEN and the anchor-sets in P-GNN are independently sampled for each epoch as You et al. (2019) does. Meanwhile, PEG is directly built upon GraphSAGE because it essentially is an approach that uses Laplacian eigenmaps to generate edge weights for the backbone model. We conduct the experiments on all datasets in two settings: 1. with node attributes and 2. without node attributes. In the case of the setting without node attributes, we adopt the approach used in Yin et al. (2020), where the degrees of each node replace the node attributes as the input.

We follow the training procedure proposed by Pei et al. (2020) for all experiments. Models are trained and evaluated with ten fixed random seeds. We use splits of training, validation, and test sets provided by Pei et al. (2020) on Cora and Citeseer. The rest of the datasets come with splits provided by Platonov et al. (2023). We measure the performance of Minesweeper and Tolokers using ROC AUC and calculate accuracy for the rest of the datasets. The accuracy or ROC AUC on the test set is recorded when the model achieves the highest score on the validation set. All results are reported as an average of ten fixed splits of the test set. The Adam optimizer is adopted with $\beta_1 = 0.9$, and $\beta_2 = 0.999$. The temperature parameter $\tau$ used in InfoNCE loss is set to 1. During the training, the learning rate is reduced by a factor of two after ten epochs without improvement of the loss. The training progress will stop when the learning rate is lower than $1e-7$ or the number of epochs exceeds 200. We randomly select about $0.005\% - 30\%$ of the nodes across a graph with equal probabilities to form the referential node set. Other hyper-parameters are searched through grid search and selected according to the best evaluation set results. The appendix A lists the search space, all final hyper-parameters, and model architectures. The code is implemented using PyTorch Amos et al. (2018) and runs on an Nvidia H100 94GB GPU.

## 4.3 Results and Analysis

We compared GPEN with other state-of-the-art encoding methods in situations where node attributes are present or absent. We only performed experiments using node attributes in Minesweeper because its label informativeness LI is zero, meaning that structural information is meaningless without incorporating node attributes. We generally find that GraphSAGE-based methods perform better than NAGphormer-based methods. PE methods, especially GPEN, outperform DE methods on graphs with positive adjusted homophily $h_{adj}$ and LI when the node attribute is absent. The more graphs tend to be homophilic, the larger

Table 2: The performance of state-of-the-art and GPEN augmented models on datasets whose adjusted homophily $h_{adj} \geq 0.68$. Accuracy (%) is reported. Red: the best results. Orange: the second-best results. Blue: the third-best results.

| Type | Dataset $h_{adj}$ LI Attr. | Cora 0.77 0.59 | | Cite. 0.68 0.46 | |
|---|---|---|---|---|---|
| | | w/ | w/o | w/ | w/o |
| None | GraphSAGE | 86.12 ± 0.96 | 41.41 ± 2.54 | 76.12 ± 1.88 | 33.26 ± 1.73 |
| DE | GraphSAGE-DE-GNN | 86.30 ± 1.72 | 41.01 ± 1.29 | 76.45 ± 1.92 | 24.25 ± 2.38 |
| | GraphSAGE-PEG | 85.09 ± 1.72 | 41.45 ± 2.27 | 76.58 ± 1.79 | 31.14 ± 2.77 |
| PE | GraphSAGE-P-GNN | 85.33 ± 1.91 | 59.78 ± 2.38 | 75.03 ± 2.32 | 46.68 ± 5.26 |
| | GraphSAGE-GPEN (Ours) | 86.34 ± 1.36 | 75.27 ± 2.05 | 76.19 ± 1.56 | 51.49 ± 5.54 |
| | NAGphormer | 83.72 ± 2.04 | 52.60 ± 2.14 | 74.15 ± 2.27 | 50.07 ± 5.37 |
| | NAGphormer-P-GNN | 83.78 ± 1.66 | 39.48 ± 6.16 | 74.10 ± 2.42 | 44.23 ± 5.34 |
| | NAGphormer-GPEN (Ours) | 84.08 ± 1.86 | 77.20 ± 2.09 | 74.33 ± 1.44 | 51.15 ± 5.95 |

Table 3: The performance of state-of-the-art and GPEN augmented models on datasets whose adjusted homophily $h_{adj} < 0.68$. Accuracy (%) is reported for Roman-empire and Amazon-ratings. ROC AUC (%) is reported for Minesweeper and Tolokers. Red: the best results. Orange: the second-best results. Blue: the third-best results.

| Type | Dataset $h_{adj}$ LI Attr. | Roma. -0.05 0.11 | | Amazon. 0.14 0.04 | | Mine. 0.01 0.00 | Tolokers 0.09 0.01 | |
|---|---|---|---|---|---|---|---|---|
| | | w/ | w/o | w/ | w/o | w/ | w/ | w/o |
| None | GraphSAGE | 81.73 ± 0.50 | 30.67 ± 0.79 | 49.76 ± 0.41 | 36.97 ± 0.31 | 90.82 ± 0.62 | 83.54 ± 0.70 | 69.92 ± 1.58 |
| DE | GraphSAGE-DE-GNN | 82.12 ± 0.43 | 36.42 ± 0.60 | 50.06 ± 0.77 | 37.65 ± 0.54 | 90.93 ± 0.79 | 83.48 ± 0.84 | 68.82 ± 1.00 |
| | GraphSAGE-PEG | 81.66 ± 0.64 | 31.84 ± 0.49 | 49.45 ± 0.65 | 36.99 ± 0.24 | 90.86 ± 0.64 | 83.26 ± 0.70 | 68.75 ± 1.21 |
| PE | GraphSAGE-P-GNN | 82.28 ± 0.28 | 32.05 ± 0.28 | 50.05 ± 0.49 | 41.88 ± 0.69 | 90.59 ± 0.20 | 83.72 ± 0.35 | 75.53 ± 0.58 |
| | GraphSAGE-GPEN (Ours) | 83.79 ± 0.69 | 36.39 ± 0.44 | 49.42 ± 0.32 | 38.41 ± 0.99 | 90.83 ± 0.73 | 84.42 ± 0.79 | 79.77 ± 0.96 |
| | NAGphormer | 77.02 ± 0.78 | 21.08 ± 0.47 | 44.10 ± 0.66 | 42.26 ± 0.52 | 86.22 ± 1.40 | 82.52 ± 1.04 | 74.39 ± 0.90 |
| | NAGphormer-P-GNN | 77.37 ± 0.78 | 22.13 ± 1.11 | 43.63 ± 0.84 | 38.11 ± 1.99 | 87.40 ± 1.46 | 82.86 ± 1.16 | 74.22 ± 2.07 |
| | NAGphormer-GPEN (Ours) | 80.12 ± 0.48 | 34.56 ± 0.50 | 43.25 ± 0.60 | 37.87 ± 1.25 | 86.79 ± 1.16 | 84.38 ± 0.92 | 79.68 ± 1.14 |

the margin. Thus, we roughly divide the results into Table 2 and 3 for datasets with $h_{adj} \geq 0.68$ and $h_{adj} < 0.68$, respectively.

When $h_{adj} \geq 0.68$, models typically benefit from the inherent global structure where similar nodes are more likely to be connected. As a result, PEs of nodes with the same or different labels can be similar or different, which poses a challenge for DE methods to provide such information. In this context, GPEN-augmented GraphSAGE models demonstrated substantial performance improvements, particularly when unavailable node attributes. For instance, on the Cora dataset, GraphSAGE-GPEN elevates the accuracy of the backbone by 33.86% without attributes, a dramatic improvement that showcases GPEN's ability to use global information effectively. Similarly, on the Citeseer dataset, GPEN improves the backbone model's performance without attributes from 33.26% to 51.49%, underlining its robustness in leveraging the graph's intrinsic properties. Moreover, our GPEN surpasses other PE methods, such as P-GNN and Laplacian eigenmaps, with 1.08% (51.15% vs. 50.07%) to 37.72% (77.20% vs. 39.48%) differences. Although GPEN is not always ahead of other DE methods when using node attributes, the performance differences are minor, and GPEN can still boost the backbone models' performance.

In scenarios where $h_{adj} < 0.68$, graphs are more challenging for models due to the weak correlation between node connectivity and labels. Besides node attributes, the heterophilic graphs demand more structural information that distinguishes subtle and complex structural patterns to predict node labels effectively. Despite solving these challenges, PE and DE methods demonstrate varied performance, illustrating that their efficacy can be context-dependent.

For example, on the Roman-empire dataset, which has the lowest adjusted homophily score and where node attributes already offer high predictive utility, GPEN can still improve the accuracy of the GraphSAGE model by 2.09%, much higher than other datasets in the same setting. This demonstrates GPEN's ability to capture useful structural features even in complex graph environments. Moreover, GPEN-augmented models

consistently outperform other DE and PE methods on Tolokers with and without using node attributes, even when the graph is heterophilic and has extremely low label informativeness.

However, it is crucial to note that in specific contexts like the Amazon-ratings dataset, GPEN does not perform as well, with state-of-the-art methods like DE-GNN and P-GNN outperforming GPEN. Meanwhile, unlike DE methods, the GPEN-augmented GraphSAGE model only achieves third-place performance with a ROC AUC of 90.83% in the Minesweeper dataset.

Considering the rank of performances when comparing all GraphSAGE-based models, GPEN achieves six times the best results out of 11 settings, which is significant better than other encoding methods. Thus, GPEN is still a superior PE method to extract global structural information in graphs, despite the various performance.

### 4.4 Ablation Study

### 4.4.1 Stability of Global Positional Embeddings

It is essential to examine the stability of the global positional embeddings generated by GPEN when the referential nodes change, as they are randomly selected. Therefore, we evaluate the stability of pre-trained GPEN, and GPEN without using InfoNCE loss (GPEN-w/o-$l_{NCE}$) across all datasets with GraphSAGE backbone model. We randomly select the referential nodes 1,000 times for each test set split. We calculate the average standard deviations (s.d.) of the accuracy or ROC AUC for each split to obtain the s.d. result of the dataset, as shown in Table 4 and 5.

Table 4: Ablation study on stability of global positional embeddings. Models are trained on datasets whose adjusted homophily $h_{adj} \geq 0.68$. The standard deviations (s.d.) of accuracy (%) are compared in parallel. A lower s.d. is better. ↑: higher s.d. of GPEN after using the InfoNCE loss. ↓: lower s.d. of GPEN after using the InfoNCE loss.

| Dataset | Cora | | Cite. | |
| --- | --- | --- | --- | --- |
| Attr. | w/ | w/o | w/ | w/o |
| GraphSAGE-GPEN-w/o-$l_{NCE}$ (Ours) | 0.428 | 1.419 | 0.514 | 1.047 |
| GraphSAGE-GPEN (Ours) | ↓0.419 | ↓1.384 | ↑0.551 | ↓1.026 |

Table 5: Ablation study on stability of global positional embeddings. Models are trained on datasets whose adjusted homophily $h_{adj} < 0.68$. The standard deviations (s.d.) of accuracy (%) or ROC AUC (%) are compared in parallel. A lower s.d. is better. ↑: higher s.d. of GPEN after using the InfoNCE loss. ↓: lower s.d. of GPEN after using the InfoNCE loss.

| Dataset | Roma. | | Amazon. | | Mine. | Tolokers | |
| --- | --- | --- | --- | --- | --- | --- | --- |
| Attr. | w/ | w/o | w/ | w/o | w/ | w/ | w/o |
| GraphSAGE-GPEN-w/o-$l_{NCE}$ (Ours) | 0.128 | 0.312 | 0.030 | 0.318 | 0.004 | 0.668 | 1.286 |
| GraphSAGE-GPEN (Ours) | ↑0.152 | ↓0.295 | ↑0.099 | ↓0.313 | ↑0.018 | ↓0.584 | ↓1.051 |

According to the findings, even using the simple strategy for sampling referential nodes, GPEN still keeps a relatively low s.d. compared to the results on all datasets. Meanwhile, using the InfoNCE loss generally helps enhance GPEN's stability, with seven out of 11 settings highlighting the importance of incorporating contrastive learning techniques. Compared to the P-GNN sampling anchor-set based on the Bourgain theorem Bourgain (1985), our philosophy is to trade off the sampling complexity to achieve flexibility while maintaining higher accuracy by a notable margin using a simple sampling strategy and contrastive learning.

### 4.4.2 Transfer learning of GPEN

We conduct transfer learning experiments to test GPEN's ability to learn graph structure knowledge. We load and freeze the weights of the GPEN module in the pre-trained GraphSAGE-GPEN on the source dataset and only train the backbone model with cross-entropy loss on the target dataset from scratch. We denote such model as GraphSAGE-GPEN-*source dataset*. We follow the same training procedure used in

Sec. 4.2, including searching the size of referential node set. Here, we select Cora and Roman-empire as the source or target dataset. Table 6 shows that GPEN's performance is significantly downgraded after being transferred to the new domain when not using node attribute. However, a new model that uses transferred GPEN can achieve better results than directly training the whole model on the target dataset (Cora to Roma.) when node attributes are present. It can still outperform P-GNN, even suffering a slight downgrade in the Roma. to Cora setting using node attributes. The results suggest that GPEN can generalize the graph structure knowledge and potentially become a pretrained model to provide positional information to downstream networks if trained on various graphs on a large scale. Meanwhile, training GPEN jointly with node attributes might be significant in achieving such a goal.

Table 6: Ablation study of transfer learning of GPEN. Accuracy (%) is compared in parallel.

| Transfer learning | Dataset | Cora | | Roma. | |
| | Attr. | w/ | w/o | w/ | w/o |
| --- | --- | --- | --- | --- | --- |
| ✗ | GraphSAGE-P-GNN | $85.33 \pm 1.91$ | $59.78 \pm 2.38$ | $82.28 \pm 0.28$ | $32.05 \pm 0.28$ |
| | GraphSAGE-GPEN (Ours) | $86.34 \pm 1.36$ | $75.27 \pm 2.05$ | $83.79 \pm 0.69$ | $36.39 \pm 0.44$ |
| ✓ | GraphSAGE-GPEN-*Roma.* (Ours) | $86.26 \pm 1.25$ | $53.20 \pm 2.88$ | - | - |
| | GraphSAGE-GPEN-*Cora* (Ours) | - | - | $84.70 \pm 0.46$ | $33.12 \pm 0.39$ |

### 4.5 Limitations

Table 7: Prepossessing computational complexity and spatial complexity after prepossessing of GPEN and Laplacian eigenmaps. $|\mathbb{V}|$: the number of nodes. $k$: the number of steps of random walk. $|\mathbb{O}|$: the size of referential node set. $p$: $p$ smallest eigenvectors.

| | Prepossessing computational complexity | Spatial complexity after prepossessing |
| --- | --- | --- |
| GPEN | $\mathcal{O}(|\mathbb{V}|^3 k)$ | $\mathcal{O}(|\mathbb{V}| |\mathbb{O}| k)$ |
| Laplacian eigenmaps | $\mathcal{O}(|\mathbb{V}|^3)$ | $\mathcal{O}(|\mathbb{V}| p)$ |

The main limitation of GPEN is that we use random walk probabilities as the inputs to calculate global positional embeddings. As Table 7 shows, GPEN requires a longer time for prepossessing than Laplacian eigenmaps. Meanwhile, GPEN also uses significant memory to store inputs after prepossessing. Training on large graphs is challenging when the referential node size is large. Thus, we suggest exploring lightweight algorithms, such as the shortest path, to generate inputs for GPEN in future works.

## 5 Conclusion

We have proposed GPEN, which generates global positional embeddings for all nodes in a graph without needing either node or edge attributes. This enhances discrimination with a relatively high degree of consistency. Our experiments have demonstrated that our GPEN is superior to DE methods in representing structural information on various graphs with different homophily. Our GPEN can be integrated with any GNN approach to effectively enhance their performance, particularly on node classification tasks on graphs without node attributes. We demonstrate the potential of GPEN to be trained on various graph structures to learn more general global positional embeddings, which can then be applied to multiple tasks.

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

## A   Appendix

### A.1   Code

Our code[1] is implemented using PyTorch Amos et al. (2018), DGL Wang et al. (2019), and following projects:

1. Benchmarking Graph Neural Networks[2] Dwivedi et al. (2020);

2. Geo-GCN[3] Pei et al. (2020);

3. DE-GNN[4] Yin et al. (2020);

4. PEG[5] Wang et al. (2022);

5. P-GNN[6] You et al. (2019);

6. SimCLR[7] Chen et al. (2020).

7. Heterophilous graphs[8] Platonov et al. (2023)

8. NAGphormer[9] Chen et al. (2023)

### A.2   Architecture Details

The details of the models used in this paper are illustrated in the following section.

### A.2.1   GraphSAGE Backbone

The GraphSAGE Hamilton et al. (2017) module in the *l*-th layer of GraphSAGE backbone is formulated as:

$$\boldsymbol{h}_v^{l+1} = \mathrm{ReLU}\big(\boldsymbol{U}\mathrm{Concat}(\boldsymbol{h}_v^l, \mathrm{Mean}_{u \in \mathcal{N}(v)}\boldsymbol{h}_u^l)\big), \tag{14}$$

where $\boldsymbol{U} \in \mathbb{R}^{d_{out} \times 2d_{in}}$, $\mathcal{N}(v)$ is the neighbours of the central node $v$. The architecture of the GraphSAGE backbone used for the node classification task in this paper is illustrated in Fig. 4.

---

[1]The source code is available at `https://github.com/Anonymous/GPEN`

[2]`https://github.com/graphdeeplearning/benchmarking-gnns`

[3]`https://github.com/graphdml-uiuc-jlu/geom-gcn`

[4]`https://github.com/VeritasYin/DEGNN_node_classification`

[5]`https://github.com/Graph-COM/PEG`

[6]`https://github.com/RecLusIve-F/P-GNN-dgl`

[7]`https://github.com/Spijkervet/SimCLR`

[8]`https://github.com/yandex-research/heterophilous-graphs`

[9]`https://github.com/JHL-HUST/NAGphormer`

Figure 4: Architecture of the GraphSAGE backbone. $\boldsymbol{h}_v$: the node attribute of the central node $v$. $\{\boldsymbol{h}_{\mathcal{N}(v)}\}$ the node attribute set of the central node $v$'s neighbours. $d_{in}$: the dimension of inputs. $d_{hid}$: the dimension of hidden layers. $d_{class}$: the number of classes. $\boldsymbol{U}_{class1} \in \mathbb{R}^{d_{hid} \times d_{hid}}$. $\boldsymbol{U}_{class2} \in \mathbb{R}^{d_{class} \times d_{hid}}$.

### A.2.2 NAGphormer Backbone

The NAGphormer Chen et al. (2023) designs a Hop2Token module that uses parameter-free functions to aggregate the neighbourhood features from different hops and tokenizes them as a sequence. Given the node attribute matrix $\boldsymbol{H} \in \mathbb{R}^{|\mathbb{V}| \times d_{in}}$, $p$ smallest Laplacian eigenvector matrix $\boldsymbol{Z} \in \mathbb{R}^{|\mathbb{V}| \times p}$, the $i$th-hop neighborhood matrix is formulated as:

$$\boldsymbol{H}_i = \begin{cases} \text{Concat}(\boldsymbol{H}, \boldsymbol{Z}), & i = 0, \\ \boldsymbol{W}\text{Concat}(\boldsymbol{H}, \boldsymbol{Z}), & i = 1, 2, ..., n_{tok}, \end{cases} \tag{15}$$

where $\text{Concat}(\cdot, \cdot)$ is concatenation alone the feature dimension. $n_{tok}$ is the number of tokenized hops, $\boldsymbol{W}$ is the transition matrix. We denote the sequence tensor as $\mathsf{S} = (\boldsymbol{H}_1, \boldsymbol{H}_2, ..., \boldsymbol{H}_{n_{tok}}) \in \mathbb{R}^{|\mathbb{V}| \times (n_{tok}+1) \times (d_{in}+p)}$.

Following an embedding layer composed of $\boldsymbol{U}_{emb} \in \mathbb{R}^{(d_{in}+p) \times d_{hid}}$, the sequence tensor $\mathsf{S}$ is embedded into a $d_{hid}$-dimension feature space as $\mathsf{S}^{(0)} \in \mathbb{R}^{|\mathbb{V}| \times (n_{tok}+1) \times d_{hid}}$ that is the input of Transformer encoder.

NAGphormer uses Transformer encoders proposed by Vaswani et al. (2017) to encode each node's sequential features. Given a target node $v$, its feature of the $l$-layer of Transformer encoder is formulated as:

$$\boldsymbol{S}_v^{'(l)} = \text{MSA}\big(\text{LN}(\boldsymbol{S}_v^{(l-1)})\big) + \boldsymbol{S}_v^{(l-1)}, \tag{16}$$

$$\boldsymbol{S}_v^{(l)} = \text{FFN}\big(\text{LN}(\boldsymbol{S}_v^{'(l)})\big) + \boldsymbol{S}_v^{'(l)}, \tag{17}$$

where $\boldsymbol{S}_v^{(l)} \in \mathbb{R}^{(n_{tok}+1) \times d_{hid}}$. MSA is multi-head self-attention. FFN is position-wise feed-forward network. LN is LayerNorm.

Ultimately, NAGphormer proposes an attention-based readout function for node classification tasks. The output of the last Transformer encoder layer $\boldsymbol{S}_v \in \mathbb{R}^{(n_{tok}+1)}$ is decomposed as the token representation of the node itself $\boldsymbol{F}_0 \in \mathbb{R}^{d_{hid}}$ and its $i$th hop representation $\boldsymbol{F}_i \in \mathbb{R}^{d_{hid}}$. The output feature vector of $v$ is formulated as:

$$\boldsymbol{F}_{out} = \boldsymbol{F}_0 + \sum_{i=1}^{n_{head}} \alpha_i \boldsymbol{F}_i, \tag{18}$$

$$\alpha_i = \frac{e^{\text{Concat}(\boldsymbol{F}_0, \boldsymbol{F}_i)\boldsymbol{U}_a}}{\sum_{j=1}^{n_{head}} e^{\text{Concat}(\boldsymbol{F}_0, \boldsymbol{F}_j)\boldsymbol{U}_a}}, \tag{19}$$

where $\boldsymbol{U}_a \in \mathbb{R}^{2d_{hid} \times 1}$. $n_{head}$ is the number of attention heads.

The architecture of the NAGphormer backbone used for the node classification task in this paper is illustrated in Fig. 5. Because NAGphormer originally uses Laplacian eigenvectors as the positional embeddings, replacing it with other PE methods is straightforward when using NAGphormer as the backbone model.

### A.2.3 DE-GNN

The DE-GNN Li et al. (2020) generates a distance embedding for each node pair using $k$-hop random walk distance. Given a central node $v$ and a node $u$, the embedding $\boldsymbol{l}_{uv} \in \mathbb{R}^k$ is formulated as:

$$\boldsymbol{l}_{uv} = [(\boldsymbol{W})_{uv}, (\boldsymbol{W}^2)_{uv}, ..., (\boldsymbol{W}^k)_{uv}], \tag{20}$$

where $(\boldsymbol{W}^k)_{uv}$ is the $k$-step random walk probability from the node $u$ to $v$. The architecture of the DE-GNN with the GraphSAGE backbone used for the node classification task in this paper is illustrated in Fig. 6.

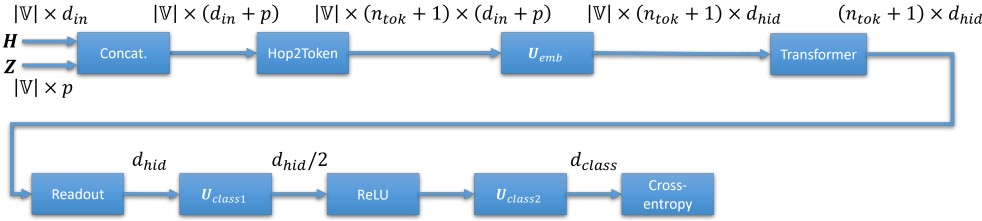

Figure 5: Architecture of the NAGphormer backbone. $\boldsymbol{H}$: the node attribute of all nodes in the graph. $\boldsymbol{Z}$: the $p$ smallest Laplacian eigenvectors of all nodes in the graph. $\mathbb{V}$: the node set of the graph. $d_{in}$: the dimension of inputs. $n_{tok}$: the number of tokenized hops. $d_{hid}$: the dimension of hidden layers. $d_{class}$: the number of classes. $\boldsymbol{U}_{class1} \in \mathbb{R}^{d_{hid}/2 \times d_{hid}}$. $\boldsymbol{U}_{class2} \in \mathbb{R}^{d_{class}/2 \times d_{hid}}$.

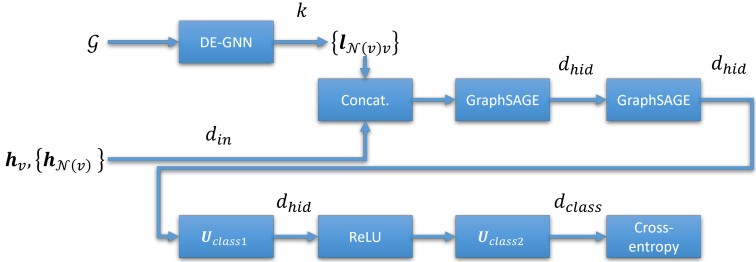

Figure 6: Architecture of the DE-GNN combined with the GrapSAGE backbone. $\mathcal{G}$: the input graph. $\boldsymbol{h}_v$: the node attribute of the central node $v$. $\{\boldsymbol{h}_{\mathcal{N}(v)}\}$ the node attribute set of the central node $v$'s neighbours. $d_{in}$: the dimension of inputs. $k$: the number of hops of random walk. $\{\boldsymbol{l}_{\mathcal{N}(v)v}\}$: the set of distance embeddings from the central node $v$'s neighbours to the $v$. $d_{hid}$: the dimension of hidden layers. $d_{class}$: the number of classes. $\boldsymbol{U}_{class1} \in \mathbb{R}^{d_{hid} \times d_{hid}}$. $\boldsymbol{U}_{class2} \in \mathbb{R}^{d_{class} \times d_{hid}}$.

### A.2.4  PEG

The PEG Wang et al. (2022) aims to generate a stable edge weight using Laplacian eigenmaps:

$$\xi_{uv} = \sigma\big(\boldsymbol{U}_2 \boldsymbol{U}_1 \|\boldsymbol{z}_u - \boldsymbol{z}_v\|\big), \tag{21}$$

where $\boldsymbol{z}_u$ and $\boldsymbol{z}_v$ are $p$ smallest eigenvectors of the node $u$ and $v$, respectively. $\boldsymbol{U}_1^T, \boldsymbol{U}_2 \in \mathbb{R}^{d_{hid_e}}$.

When combining the PEG with the GraphSAGE module, Eq. 14 is re-formulated as:

$$\boldsymbol{h}_v^{l+1} = \mathrm{ReLU}\Big(\boldsymbol{U}\,\mathrm{Concat}\big(\boldsymbol{h}_v^l, \mathrm{Mean}_{u\in\mathcal{N}(v)}(\xi_{uv}\boldsymbol{h}_u^l)\big)\Big). \tag{22}$$

The architecture of the PEG combined with the GraphSAGE backbone used for the node classification task in this paper is illustrated in Fig. 7.

### A.2.5  P-GNN

Given any two nodes $v$ and $u$ in a graph $\mathcal{G} = (\mathbb{V}, \mathbb{E})$, P-GNN You et al. (2019) first defines their distance in the $\mathcal{G}$:

$$s(v, u) = \frac{1}{d_{sp}^q(v, u) + 1}, \tag{23}$$

$$d_{sp}^q(v, u) = \begin{cases} d_{sp}(v, u), & \text{if } d_{sp}(v, u) \leq q, \\ \infty, & \text{otherwise}, \end{cases} \tag{24}$$

where $d_{sp}(v, u)$ is the shortest path between $v$ and $u$. $q$ is the longest searching hop which should be no larger than the diameter of $\mathcal{G}$.

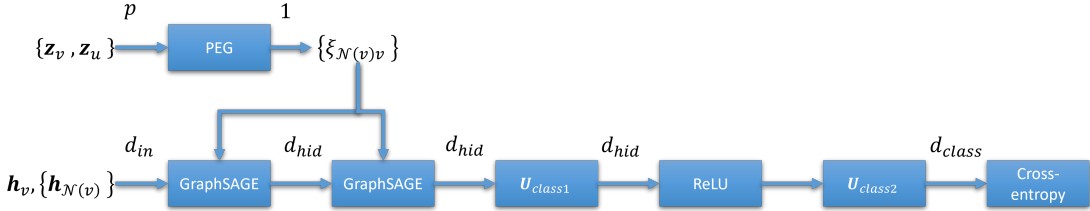

Figure 7: Architecture of the PEG combined with the GraphSAGE backbone. $\boldsymbol{h}_v$: the node attribute of the central node $v$. $\{\boldsymbol{h}_{\mathcal{N}(v)}\}$ the node attribute set of the central node $v$'s neighbours. $\boldsymbol{z}_{\mathcal{N}}(v)$: $p$ smallest eigenvectors of the central node $v$'s neighbours. $\boldsymbol{z}_v$: $p$ smallest eigenvectors of the central node $v$. $\{\xi_{\mathcal{N}(v)v}\}$: the set of edge weights between the central node $v$'s neighbours and the $v$. $d_{in}$: the dimension of inputs. $d_{hid}$: the dimension of hidden layers. $d_{class}$: the number of classes. $\boldsymbol{U}_{class1} \in \mathbb{R}^{d_{hid} \times d_{hid}}$. $\boldsymbol{U}_{class2} \in \mathbb{R}^{d_{class} \times d_{hid}}$.

Following the Bourgain theorem Bourgain (1985), P-GNN samples anchor-sets $\mathbb{S}_{i,j} \subset \mathbb{V}, i = 1, 2, ..., \lceil \log n \rceil, j = 1, 2, ..., \lceil \log n \rceil, n = |\mathbb{V}|$. The total number of anchor-sets is $N_{\mathbb{S}} = \lceil \log n \rceil^2$. For each anchor-set $\mathbb{S}_{i,j}$, P-GNN independently sample each node from $\mathbb{V}$ with probability $\frac{1}{2^i}$.

For each node $v \in \mathbb{V}$ in the $l$-th layer of the P-GNN module, a matrix $\boldsymbol{M}^l \in \mathbb{R}^{d_{hid} \times N_{\mathbb{S}}}$ of anchor-set messages is generated based on anchor-sets. Each column of $\boldsymbol{M}^l$ is an anchor-set message $\boldsymbol{M}_i^l$ formulated as:

$$\boldsymbol{M}_i^l = \text{ReLU}\big(s(v, u)\boldsymbol{U}_1^l\text{Concat}(\boldsymbol{h}_v^l, \boldsymbol{h}_u^l)\big), \forall u \in \underset{u \in \mathbb{S}_{i,j}}{\arg\min}\, d(v, u), i = 1, 2, ..., \lceil \log n \rceil, j = 1, 2, ..., \lceil \log n \rceil, \quad (25)$$

where $\boldsymbol{U}_1^l \in \mathbb{R}^{d_{out} \times 2d_{in}}$. Then, we can obtain the position-aware embedding $\boldsymbol{z}_v^l$ and the message $\boldsymbol{h}_v^{l+1}$ for the node $v$:

$$\boldsymbol{z}_v^l = \boldsymbol{U}_2^l\boldsymbol{M}^l, \quad (26)$$

$$\boldsymbol{h}_v^{l+1} = \text{Mean}_{i \in \{1,2,...,N_{\mathbb{S}}\}}\boldsymbol{M}_i^l, \quad (27)$$

where $\boldsymbol{U}_2^l \in \mathbb{R}^{d_{out}}$.

Following the original paper, we use two layers of the P-GNN module. The architecture of the P-GNN combined with the GraphSAGE backbone used for the node classification task in this paper is illustrated in Fig. 8.

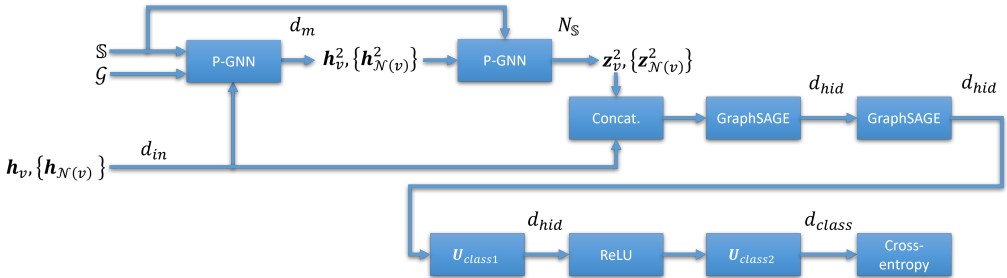

Figure 8: Architecture of the P-GNN combined with the GraphSAGE backbone. $\boldsymbol{h}_v$: the node attribute of the central node $v$. $\{\boldsymbol{h}_{\mathcal{N}(v)}\}$ the node attribute set of the central node $v$'s neighbours. $\boldsymbol{h}_v^2$: the second layer input message of the central node $v$. $\boldsymbol{h}_{\mathcal{N}}^2(v)$: the set of the second layer input messages of the central node $v$'s neighbours. $d_m$: the dimension of the messages. $\boldsymbol{z}_v^2$: the second layer output position-aware embedding of the node $v$. $\{\boldsymbol{z}_{\mathcal{N}}^2(v)\}$: the set of the second layer output position-aware embeddings of the central node $v$'s neighbours. $d_{in}$: the dimension of inputs. $d_m$: the dimension of output messages in the P-GNN. $N_{\mathbb{S}}$: the total number of anchor-sets. $d_{hid}$: the dimension of hidden layers. $d_{class}$: the number of classes. $\boldsymbol{U}_{class1} \in \mathbb{R}^{d_{hid} \times d_{hid}}$. $\boldsymbol{U}_{class2} \in \mathbb{R}^{d_{class} \times d_{hid}}$.

### A.2.6 GPEN

As Fig. 9 shows, our GPEN consists of two two-layer MLPs and summation aggregation to generate the global positional embeddings for each node. Fig. 10 illustrates the projection head that maps the global positional embeddings to a representation space where the InfoNCE loss Oord et al. (2018) trains the GPEN. The architecture of the GPEN combined with the GraphSAGE backbone used for the node classification task in this paper is shown in Fig. 11.

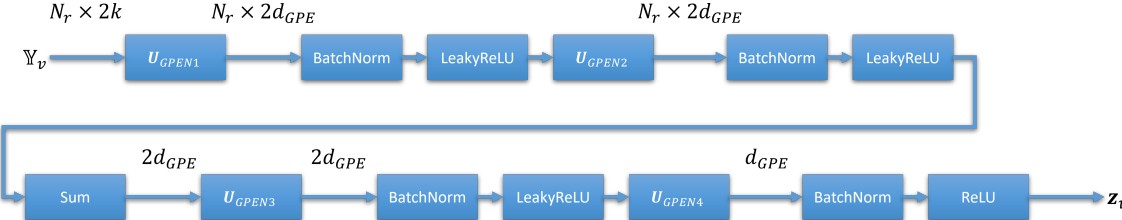

Figure 9: Architecture of the GPEN. $k$: the number of steps of random walk. $\mathbb{Y}_v$: The $k$-step random walk probability vector set representing distances from referential nodes to node $v$. $N_r$: the number of referential nodes. $d_{GPE}$: the dimension of the global positional embedding. $z_v$: the global positional embedding of the node $v$. $U_{GPEN1} \in \mathbb{R}^{2k \times 2d_{GPE}}$. $U_{GPEN2}, U_{GPEN3} \in \mathbb{R}^{2d_{GPE} \times 2d_{GPE}}$. $U_{GPEN4} \in \mathbb{R}^{2d_{GPE} \times d_{GPE}}$

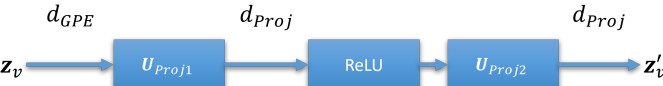

Figure 10: Architecture of projection head. $z_v$: the global positional embedding of the node $v$. $d_{GPE}$: the dimension of the global positional embedding. $d_{Proj}$: the output dimension of the projection head. $z'_v$: the projection of the node $v$'s global positional embedding. $U_{Proj1} \in \mathbb{R}^{d_{GPE} \times d_{Proj}}$. $U_{Proj2} \in \mathbb{R}^{d_{Proj} \times d_{Proj}}$.

### A.3 Hyper-parameter Details

The hyper-parameter search space of all experiments conducted by ourselves is listed as Table 8. After the grid search, we select the final hyper-parameters for each experiment according to its best validation set result. The final hyper-parameters of each experiment are listed as Table 9, 10, 11, 12, 13, 14, 15, 16, 17, 18, 19, 20, 21, 22, 23, 24, 25, 26, 27, 28, 29., and 30.

Table 8: Hyper-parameter search space of all experiments conducted by ourselves. $k$: the number of steps of random walk. $N_r$ factor: percentage of referential nodes compared to total nodes in a graph. $lr_{init}$: initial learning rate. $bs$: batch size.

| Hyperparameter | Search space | Dataset |
|---|---|---|
| $k$ | 5, 10, 20 | All |
| $Nr$ factor | 0.1%, 0.5%, 1%, 10%, 20%, 30%
0.1%, 0.5%, 1%, 10%, 20% | Cora, Cite., Roma., Mine., Tolokers
Amazon. |
| $lr_{init}$ | 1e-4, 5e-4, 1e-3, 5e-3 | All |
| $bs$ | 16, 32, 64 | All |

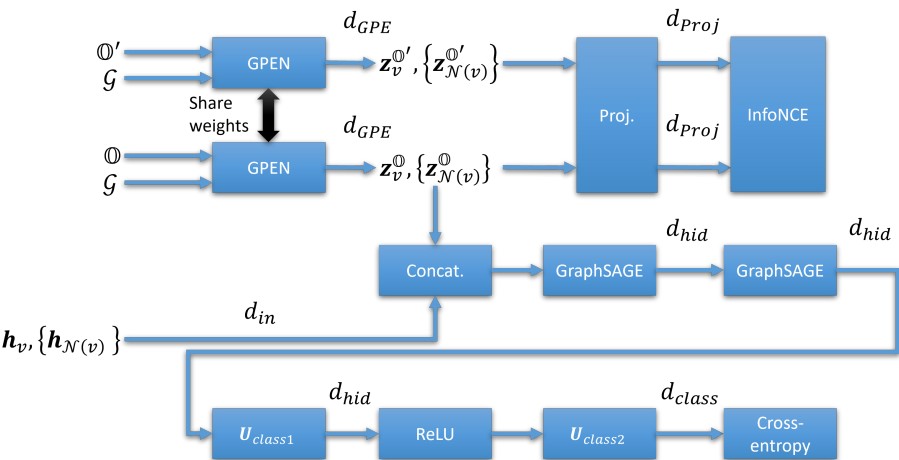

Figure 11: Architecture of the GPEN combined with the GrapSAGE backbone. $\mathbb{O}$, $\mathbb{O}'$: the referential node sets with different node selections. $\mathcal{G}$: the input graph. $\boldsymbol{h}_v$: the node attribute of the central node $v$. $\{\boldsymbol{h}_{\mathcal{N}(v)}\}$ the node attribute set of the central node $v$'s neighbours. $d_{in}$: the dimension of inputs. $d_{GPE}$: the dimension of the global positional embedding. $\boldsymbol{z}_v^{\mathbb{O}}$, $\boldsymbol{z}_v^{\mathbb{O}'}$: the global positional embedding of the node $v$ under the corresponding referential node set. $\{\boldsymbol{z}_{\mathcal{N}}^{\mathbb{O}}(v)\}$, $\{\boldsymbol{z}_{\mathcal{N}}^{\mathbb{O}'}(v)\}$: the set of the global positional embeddings of the central node $v$'s neighbours under the corresponding referential node set. $d_{Proj}$: the output dimension of the projection head. $d_{hid}$: the dimension of hidden layers. $d_{class}$: the number of classes. $\boldsymbol{U}_{class1} \in \mathbb{R}^{d_{hid} \times d_{hid}}$. $\boldsymbol{U}_{class2} \in \mathbb{R}^{d_{class} \times d_{hid}}$.

Table 9: Final hyper-parameters of the GraphSAGE using node attributes. $lr_{init}$: initial learning rate. $wd$: weight decay. $bs$: batch size. $drop$: dropout rate. $d_{hid}$: the number of hidden units.

| Dataset | Cora | Cite. | Roma. | Amazon. | Mine. | Tolokers |
|---|---|---|---|---|---|---|
| $lr_{init}$ | 5e-5 | 1e-4 | 1e-3 | 1e-3 | 5e-3 | 1e-3 |
| $wd$ | 1e-5 | 1e-5 | 1e-5 | 1e-5 | 1e-5 | 1e-5 |
| $bs$ | 64 | 64 | 64 | 64 | 64 | 64 |
| $drop$ | 0.1 | 0.3 | 0.2 | 0.2 | 0.2 | 0.2 |
| $d_{hid}$ | 128 | 128 | 128 | 128 | 128 | 128 |

Table 10: Final hyper-parameters of the GraphSAGE without using node attributes. $lr_{init}$: initial learning rate. $wd$: weight decay. $bs$: batch size. $drop$: dropout rate. $d_{hid}$: the number of hidden units.

| Dataset | Cora | Cite. | Roma. | Amazon. | Tolokers |
|---|---|---|---|---|---|
| $lr_{init}$ | 5e-3 | 5e-3 | 5e-4 | 1e-3 | 1e-3 |
| $wd$ | 1e-5 | 1e-5 | 1e-5 | 1e-5 | 1e-5 |
| $bs$ | 64 | 64 | 64 | 64 | 64 |
| $drop$ | 0.1 | 0.3 | 0.2 | 0.2 | 0.2 |
| $d_{hid}$ | 128 | 128 | 128 | 128 | 128 |

Table 11: Final hyper-parameters of the DE-GNN combined with the GrapSAGE backbone using node attributes. $lr_{init}$: initial learning rate. $wd$: weight decay. $bs$: batch size. $drop$: dropout rate. $k$: the number of steps of random walk. $d_{hid}$: the number of hidden units.

| Dataset | Cora | Cite. | Roma. | Amazon. | Mine. | Tolokers |
|---|---|---|---|---|---|---|
| $lr_{init}$ | 5e-5 | 5e-5 | 5e-4 | 1e-3 | 5e-3 | 1e-3 |
| $wd$ | 1e-5 | 1e-5 | 1e-5 | 1e-5 | 1e-5 | 1e-5 |
| $bs$ | 64 | 64 | 64 | 64 | 64 | 64 |
| $drop$ | 0.1 | 0.3 | 0.2 | 0.2 | 0.2 | 0.2 |
| $k$ | 5 | 5 | 20 | 5 | 20 | 5 |
| $d_{hid}$ | 128 | 128 | 128 | 128 | 128 | 128 |

Table 12: Final hyper-parameters of the DE-GNN combined with the GrapSAGE backbone without using node attributes. $lr_{init}$: initial learning rate. $wd$: weight decay. $bs$: batch size. $drop$: dropout rate. $k$: the number of steps of random walk. $d_{hid}$: the number of hidden units.

| Dataset | Cora | Cite. | Roma. | Amazon. | Tolokers |
|---|---|---|---|---|---|
| $lr_{init}$ | 1e-3 | 1e-3 | 5e-4 | 1e-3 | 1e-3 |
| $wd$ | 1e-5 | 1e-5 | 1e-5 | 1e-5 | 1e-5 |
| $bs$ | 64 | 64 | 64 | 64 | 64 |
| $drop$ | 0.1 | 0.3 | 0.2 | 0.2 | 0.2 |
| $k$ | 20 | 10 | 20 | 5 | 10 |
| $d_{hid}$ | 128 | 128 | 128 | 128 | 128 |

Table 13: Final hyper-parameters of the PEG combined with the GrapSAGE backbone using node attributes. $lr_{init}$: initial learning rate. $wd$: weight decay. $bs$: batch size. $drop$: dropout rate. $p$ smallest eigenvectors. $d_{hid_e}$: the number of hidden units in PEG. $d_{hid}$: the number of hidden units in other parts of the model.

| Dataset | Cora | Cite. | Roma. | Amazon. | Mine. | Tolokers |
|---|---|---|---|---|---|---|
| $lr_{init}$ | 1e-4 | 1e-4 | 1e-3 | 1e-3 | 5e-3 | 5e-4 |
| $wd$ | 1e-5 | 1e-5 | 1e-5 | 1e-5 | 1e-5 | 1e-5 |
| $bs$ | 64 | 64 | 64 | 64 | 64 | 64 |
| $drop$ | 0.1 | 0.3 | 0.2 | 0.2 | 0.2 | 0.2 |
| $p$ | 2 | 3 | 18 | 10 | 28 | 21 |
| $d_{hid_e}$ | 32 | 32 | 32 | 32 | 32 | 32 |
| $d_{hid}$ | 128 | 128 | 128 | 128 | 128 | 128 |

Table 14: Final hyper-parameters of the PEG combined with the GrapSAGE backbone without using node attributes. $lr_{init}$: initial learning rate. $wd$: weight decay. $bs$: batch size. $drop$: dropout rate. $p$: $p$ smallest eigenvectors. $d_{hid_e}$: the number of hidden units in PEG. $d_{hid}$: the number of hidden units in other parts of the model.

| Dataset | Cora | Cite. | Roma. | Amazon. | Tolokers |
|---|---|---|---|---|---|
| $lr_{init}$ | 5e-3 | 1e-3 | 1e-3 | 1e-3 | 5e-4 |
| $wd$ | 1e-5 | 1e-5 | 1e-5 | 1e-5 | 1e-5 |
| $bs$ | 64 | 64 | 64 | 64 | 64 |
| $drop$ | 0.1 | 0.3 | 0.2 | 0.2 | 0.2 |
| $p$ | 2 | 3 | 18 | 10 | 21 |
| $d_{hid_e}$ | 32 | 32 | 32 | 32 | 32 |
| $d_{hid}$ | 128 | 128 | 128 | 128 | 128 |

Table 15: Final hyper-parameters of the P-GNN combined with the GrapSAGE backbone using node attributes. $lr_{init}$: initial learning rate. $wd$: weight decay. $bs$: batch size. $drop$: dropout rate. $d_m$: the dimension of output messages in the P-GNN. $d_{hid}$: the number of hidden units.

| Dataset | Cora | Cite. | Roma. | Amazon. | Mine. | Tolokers |
|---|---|---|---|---|---|---|
| $lr_{init}$ | 1e-3 | 5e-5 | 1e-3 | 5e-4 | 1e-3 | 1e-3 |
| $wd$ | 1e-5 | 1e-5 | 1e-5 | 1e-5 | 1e-5 | 1e-5 |
| $bs$ | 64 | 64 | 64 | 64 | 64 | 64 |
| $drop$ | 0.1 | 0.3 | 0.2 | 0.2 | 0.2 | 0.2 |
| $d_m$ | 128 | 128 | 128 | 128 | 128 | 128 |
| $d_{hid}$ | 128 | 128 | 128 | 128 | 128 | 128 |

Table 16: Final hyper-parameters of the P-GNN combined with the GrapSAGE backbone without using node attributes. $lr_{init}$: initial learning rate. $wd$: weight decay. $bs$: batch size. $drop$: dropout rate. $d_m$: the dimension of output messages in the P-GNN. $d_{hid}$: the number of hidden units.

| Dataset | Cora | Cite. | Roma. | Amazon. | Tolokers |
|---|---|---|---|---|---|
| $lr_{init}$ | 1e-3 | 1e-3 | 1e-3 | 1e-3 | 1e-3 |
| $wd$ | 1e-5 | 1e-5 | 1e-5 | 1e-5 | 1e-5 |
| $bs$ | 64 | 64 | 64 | 64 | 64 |
| $drop$ | 0.1 | 0.3 | 0.2 | 0.2 | 0.2 |
| $d_m$ | 128 | 128 | 128 | 128 | 128 |
| $d_{hid}$ | 128 | 128 | 128 | 128 | 128 |

Table 17: Final hyper-parameters of the GPEN combined with the GrapSAGE backbone using node attributes. $N_r$ factor: percentage of referential nodes compared to total nodes in a graph. $lr_{init}$: initial learning rate. $wd$: weight decay. $bs$: batch size. $drop$: dropout rate. $k$: the number of steps of random walk. $d_{GPE}$: the dimension of the global positional embeddings. $d_{hid}$: the number of hidden units. $d_{Proj}$: the output dimension of the projection head.

| Dataset | Cora | Cite. | Roma. | Amazon. | Mine. | Tolokers |
|---|---|---|---|---|---|---|
| $N_r$ factor | 10% | 10% | 20% | 0.5% | 0.1% | 1% |
| $lr_{init}$ | 1e-3 | 1e-4 | 1e-3 | 1e-3 | 5e-3 | 1e-3 |
| $wd$ | 1e-5 | 1e-5 | 1e-5 | 1e-5 | 1e-5 | 1e-5 |
| $bs$ | 64 | 64 | 64 | 16 | 64 | 64 |
| $drop$ | 0.1 | 0.3 | 0.2 | 0.2 | 0.2 | 0.2 |
| $k$ | 5 | 5 | 5 | 5 | 10 | 5 |
| $d_{GPE}$ | 128 | 128 | 128 | 128 | 128 | 128 |
| $d_{hid}$ | 128 | 128 | 128 | 128 | 128 | 128 |
| $d_{Proj}$ | 16 | 16 | 16 | 16 | 16 | 16 |

Table 18: Final hyper-parameters of the GPEN combined with the GrapSAGE backbone without using node attributes. $N_r$ factor: percentage of referential nodes compared to total nodes in a graph. $lr_{init}$: initial learning rate. $wd$: weight decay. $bs$: batch size. $drop$: dropout rate. $k$: the number of steps of random walk. $d_{GPE}$: the dimension of the global positional embeddings. $d_{hid}$: the number of hidden units. $d_{Proj}$: the output dimension of the projection head.

| Dataset | Cora | Cite. | Roma. | Amazon. | Tolokers |
|---|---|---|---|---|---|
| $N_r$ factor | 30% | 30% | 20% | 10% | 1% |
| $lr_{init}$ | 1e-4 | 1e-3 | 1e-3 | 1e-3 | 5e-4 |
| $wd$ | 1e-5 | 1e-5 | 1e-5 | 1e-5 | 1e-5 |
| $bs$ | 64 | 64 | 64 | 32 | 64 |
| $drop$ | 0.1 | 0.3 | 0.2 | 0.2 | 0.2 |
| $k$ | 20 | 20 | 10 | 5 | 5 |
| $d_{GPE}$ | 128 | 128 | 128 | 128 | 128 |
| $d_{hid}$ | 128 | 128 | 128 | 128 | 128 |
| $d_{Proj}$ | 16 | 16 | 16 | 16 | 16 |

Table 19: Final hyper-parameters of the NAGphormer using node attributes. $lr_{init}$: initial learning rate. $wd$: weight decay. $bs$: batch size. $drop$: dropout rate. $p$: $p$ smallest eigenvectors. $n_{head}$: the number of attention heads. $n_{tok}$: the number of tokenized hops. $d_{hid}$: the number of hidden units.

| Dataset | Cora | Cite. | Roma. | Amazon. | Mine. | Tolokers |
|---|---|---|---|---|---|---|
| $lr_{init}$ | 5e-5 | 1e-4 | 5e-4 | 1e-5 | 5e-3 | 5e-4 |
| $wd$ | 1e-5 | 1e-5 | 1e-5 | 1e-5 | 1e-5 | 1e-5 |
| $bs$ | 64 | 64 | 64 | 64 | 64 | 64 |
| $drop$ | 0.1 | 0.3 | 0.2 | 0.2 | 0.2 | 0.2 |
| $p$ | 2 | 3 | 18 | 10 | 28 | 21 |
| $n_{head}$ | 8 | 8 | 8 | 8 | 8 | 8 |
| $n_{tok}$ | 2 | 2 | 2 | 2 | 2 | 2 |
| $d_{hid}$ | 128 | 128 | 128 | 128 | 128 | 128 |

Table 20: Final hyper-parameters of the NAGphormer without using node attributes. $lr_{init}$: initial learning rate. $wd$: weight decay. $bs$: batch size. $drop$: dropout rate. $p$: $p$ smallest eigenvectors. $n_{head}$: the number of attention heads. $n_{tok}$: the number of tokenized hops. $d_{hid}$: the number of hidden units.

| Dataset | Cora | Cite. | Roma. | Amazon. | Tolokers |
|---|---|---|---|---|---|
| $lr_{init}$ | 5e-5 | 1e-3 | 5e-4 | 5e-4 | 5e-4 |
| $wd$ | 1e-5 | 1e-5 | 1e-5 | 1e-5 | 1e-5 |
| $bs$ | 64 | 64 | 64 | 64 | 64 |
| $drop$ | 0.1 | 0.3 | 0.2 | 0.2 | 0.2 |
| $p$ | 2 | 3 | 18 | 10 | 21 |
| $n_{head}$ | 8 | 8 | 8 | 8 | 8 |
| $n_{tok}$ | 2 | 2 | 2 | 2 | 2 |
| $d_{hid}$ | 128 | 128 | 128 | 128 | 128 |

Table 21: Final hyper-parameters of the P-GNN combined with the NAGphormer backbone using node attributes. $lr_{init}$: initial learning rate. $wd$: weight decay. $bs$: batch size. $drop$: dropout rate. $n_{head}$: the number of attention heads. $n_{tok}$: the number of tokenized hops. $d_m$: the dimension of output messages in the P-GNN. $d_{hid}$: the number of hidden units.

| Dataset | Cora | Cite. | Roma. | Amazon. | Mine. | Tolokers |
|---------|------|-------|-------|---------|-------|----------|
| $lr_{init}$ | 1e-4 | 1e-4 | 5e-4 | 5e-4 | 5e-4 | 1e-3 |
| $wd$ | 1e-5 | 1e-5 | 1e-5 | 1e-5 | 1e-5 | 1e-5 |
| $bs$ | 64 | 64 | 64 | 64 | 64 | 64 |
| $drop$ | 0.1 | 0.3 | 0.2 | 0.2 | 0.2 | 0.2 |
| $n_{head}$ | 8 | 8 | 8 | 8 | 8 | 8 |
| $n_{tok}$ | 2 | 2 | 2 | 2 | 2 | 2 |
| $d_m$ | 128 | 128 | 128 | 128 | 128 | 128 |
| $d_{hid}$ | 128 | 128 | 128 | 128 | 128 | 128 |

Table 22: Final hyper-parameters of the P-GNN combined with the NAGphormer backbone without using node attributes. $lr_{init}$: initial learning rate. $wd$: weight decay. $bs$: batch size. $drop$: dropout rate. $n_{head}$: the number of attention heads. $n_{tok}$: the number of tokenized hops. $d_m$: the dimension of output messages in the P-GNN. $d_{hid}$: the number of hidden units.

| Dataset | Cora | Cite. | Roma. | Amazon. | Tolokers |
|---------|------|-------|-------|---------|----------|
| $lr_{init}$ | 1e-4 | 1e-3 | 5e-4 | 5e-4 | 5e-4 |
| $wd$ | 1e-5 | 1e-5 | 1e-5 | 1e-5 | 1e-5 |
| $bs$ | 64 | 64 | 64 | 64 | 64 |
| $drop$ | 0.1 | 0.3 | 0.2 | 0.2 | 0.2 |
| $n_{head}$ | 8 | 8 | 8 | 8 | 8 |
| $n_{tok}$ | 2 | 2 | 2 | 2 | 2 |
| $d_m$ | 128 | 128 | 128 | 128 | 128 |
| $d_{hid}$ | 128 | 128 | 128 | 128 | 128 |

Table 23: Final hyper-parameters of the GPEN combined with the NAGphormer backbone using node attributes. $N_r$ factor: percentage of referential nodes compared to total nodes in a graph. $lr_{init}$: initial learning rate. $wd$: weight decay. $bs$: batch size. $drop$: dropout rate. $k$: the number of steps of random walk. $d_{GPE}$: the dimension of the global positional embeddings. $d_{hid}$: the number of hidden units. $d_{Proj}$: the output dimension of the projection head.

| Dataset | Cora | Cite. | Roma. | Amazon. | Mine. | Tolokers |
|---------|------|-------|-------|---------|-------|----------|
| $N_r$ factor | 30% | 10% | 30% | 0.1% | 0.5% | 0.5% |
| $lr_{init}$ | 1e-4 | 1e-4 | 1e-4 | 5e-4 | 5e-3 | 5e-4 |
| $wd$ | 1e-5 | 1e-5 | 1e-5 | 1e-5 | 1e-5 | 1e-5 |
| $bs$ | 64 | 64 | 16 | 64 | 64 | 64 |
| $drop$ | 0.1 | 0.3 | 0.2 | 0.2 | 0.2 | 0.2 |
| $n_{head}$ | 8 | 8 | 8 | 8 | 8 | 8 |
| $n_{tok}$ | 2 | 2 | 2 | 2 | 2 | 2 |
| $k$ | 20 | 20 | 5 | 5 | 5 | 5 |
| $d_{GPE}$ | 128 | 128 | 128 | 128 | 128 | 128 |
| $d_{hid}$ | 128 | 128 | 128 | 128 | 128 | 128 |
| $d_{Proj}$ | 16 | 16 | 16 | 16 | 16 | 16 |

Table 24: Final hyper-parameters of the GPEN combined with the NAGphormer backbone without using node attributes. $N_r$ factor: percentage of referential nodes compared to total nodes in a graph. $lr_{init}$: initial learning rate. $wd$: weight decay. $bs$: batch size. $drop$: dropout rate. $n_{head}$: the number of attention heads. $n_{tok}$: the number of tokenized hops. $k$: the number of steps of random walk. $d_{GPE}$: the dimension of the global positional embeddings. $d_{hid}$: the number of hidden units. $d_{Proj}$: the output dimension of the projection head.

| Dataset | Cora | Cite. | Roma. | Amazon. | Tolokers |
|---|---|---|---|---|---|
| $N_r$ factor | 30% | 30% | 30% | 10% | 0.5% |
| $lr_{init}$ | 1e-4 | 1e-4 | 1e-4 | 5e-4 | 5e-4 |
| $wd$ | 1e-5 | 1e-5 | 1e-5 | 1e-5 | 1e-5 |
| $bs$ | 64 | 64 | 16 | 32 | 64 |
| $drop$ | 0.1 | 0.3 | 0.2 | 0.2 | 0.2 |
| $n_{head}$ | 8 | 8 | 8 | 8 | 8 |
| $n_{tok}$ | 2 | 2 | 2 | 2 | 2 |
| $k$ | 20 | 20 | 10 | 5 | 5 |
| $d_{GPE}$ | 128 | 128 | 128 | 128 | 128 |
| $d_{hid}$ | 128 | 128 | 128 | 128 | 128 |
| $d_{Proj}$ | 16 | 16 | 16 | 16 | 16 |

Table 25: Final hyper-parameters of the GPEN combined with the GrapSAGE backbone using node attributes and without using InfoNCE loss. $N_r$ factor: percentage of referential nodes compared to total nodes in a graph. $lr_{init}$: initial learning rate. $wd$: weight decay. $bs$: batch size. $drop$: dropout rate. $k$: the number of steps of random walk. $d_{GPE}$: the dimension of the global positional embedding. $d_{hid}$: the number of hidden units. $d_{Proj}$: the output dimension of the projection head.

| Dataset | Cora | Cite. | Roma. | Amazon. | Mine. | Tolokers |
|---|---|---|---|---|---|---|
| $N_r$ factor | 10% | 10% | 20% | 0.5% | 0.1% | 1% |
| $lr_{init}$ | 1e-3 | 1e-4 | 1e-3 | 1e-3 | 5e-3 | 1e-3 |
| $wd$ | 1e-5 | 1e-5 | 1e-5 | 1e-5 | 1e-5 | 1e-5 |
| $bs$ | 64 | 64 | 64 | 16 | 64 | 64 |
| $drop$ | 0.1 | 0.3 | 0.2 | 0.2 | 0.2 | 0.2 |
| $k$ | 5 | 5 | 5 | 5 | 10 | 5 |
| $d_{GPE}$ | 128 | 128 | 128 | 128 | 128 | 128 |
| $d_{hid}$ | 128 | 128 | 128 | 128 | 128 | 128 |
| $d_{Proj}$ | 16 | 16 | 16 | 16 | 16 | 16 |

Table 26: Final hyper-parameters of the GPEN combined with the GrapSAGE backbone without using node attributes and InfoNCE loss. $N_r$ factor: percentage of referential nodes compared to total nodes in a graph. $lr_{init}$: initial learning rate. $wd$: weight decay. $bs$: batch size. $drop$: dropout rate. $k$: the number of steps of random walk. $d_{GPE}$: the dimension of the global positional embedding. $d_{hid}$: the number of hidden units. $d_{Proj}$: the output dimension of the projection head.

| Dataset | Cora | Cite. | Roma. | Amazon. | Tolokers |
|---|---|---|---|---|---|
| $N_r$ factor | 30% | 30% | 20% | 10% | 1% |
| $lr_{init}$ | 1e-4 | 1e-3 | 1e-3 | 1e-3 | 5e-4 |
| $wd$ | 1e-5 | 1e-5 | 1e-5 | 1e-5 | 1e-5 |
| $bs$ | 64 | 64 | 64 | 32 | 64 |
| $drop$ | 0.1 | 0.3 | 0.2 | 0.2 | 0.2 |
| $k$ | 20 | 20 | 10 | 5 | 5 |
| $d_{GPE}$ | 128 | 128 | 128 | 128 | 128 |
| $d_{hid}$ | 128 | 128 | 128 | 128 | 128 |
| $d_{Proj}$ | 16 | 16 | 16 | 16 | 16 |

Table 27: Final hyper-parameters of the pre-trained GPEN combined with the GrapSAGE backbone using node attributes. The weights of GPEN are pre-trained on Roman-empire and frozen. The GraphSAGE backbone is trained on Cora only using cross-entropy loss. $N_r$ factor: percentage of referential nodes compared to total nodes in a graph. $lr_{init}$: initial learning rate. $wd$: weight decay. $bs$: batch size. $drop$: dropout rate. $k$: the number of steps of random walk. $d_{GPE}$: the dimension of the global positional embedding. $d_{hid}$: the number of hidden units.

| Dataset | Cora |
|---|---|
| $N_r$ factor | 30% |
| $lr_{init}$ | 1e-3 |
| $wd$ | 1e-5 |
| $bs$ | 64 |
| $drop$ | 0.1 |
| $k$ | 5 |
| $d_{GPE}$ | 128 |
| $d_{hid}$ | 128 |

Table 28: Final hyper-parameters of the pre-trained GPEN combined with the GrapSAGE backbone without using node attributes. The weights of GPEN are pre-trained on Roman-empire and frozen. The GraphSAGE backbone is trained on Cora only using cross-entropy loss. $N_r$ factor: percentage of referential nodes compared to total nodes in a graph. $lr_{init}$: initial learning rate. $wd$: weight decay. $bs$: batch size. $drop$: dropout rate. $k$: the number of steps of random walk. $d_{GPE}$: the dimension of the global positional embedding. $d_{hid}$: the number of hidden units.

| Dataset | Cora |
|---|---|
| $N_r$ factor | 20% |
| $lr_{init}$ | 1e-4 |
| $wd$ | 1e-5 |
| $bs$ | 64 |
| $drop$ | 0.1 |
| $k$ | 10 |
| $d_{GPE}$ | 128 |
| $d_{hid}$ | 128 |

Table 29: Final hyper-parameters of pre-trained GPEN combined with the GraphSAGE backbone using node attributes. The weights of GPEN are pre-trained on Cora and frozen. The GraphSAGE backbone is trained on Roman-empire only using cross-entropy loss. $N_r$ factor: percentage of referential nodes compared to total nodes in a graph. $lr_{init}$: initial learning rate. $wd$: weight decay. $bs$: batch size. $drop$: dropout rate. $k$: the number of steps of random walk. $d_{GPE}$: the dimension of the global positional embedding. $d_{hid}$: the number of hidden units.

| Dataset | Roma. |
|---|---|
| $N_r$ factor | 20% |
| $lr_{init}$ | 1e-3 |
| $wd$ | 1e-5 |
| $bs$ | 64 |
| $drop$ | 0.2 |
| $k$ | 5 |
| $d_{GPE}$ | 128 |
| $d_{hid}$ | 128 |

Table 30: Final hyper-parameters of pre-trained GPEN combined with the GraphSAGE backbone without using node attributes. The weights of GPEN are pre-trained on Cora and frozen. The GraphSAGE backbone is trained on Roman-empire only using cross-entropy loss. $N_r$ factor: percentage of referential nodes compared to total nodes in a graph. $lr_{init}$: initial learning rate. $wd$: weight decay. $bs$: batch size. $drop$: dropout rate. $k$: the number of steps of random walk. $d_{GPE}$: the dimension of the global positional embedding. $d_{hid}$: the number of hidden units.

| Dataset | Roma. |
|---|---|
| $N_r$ factor | 30% |
| $lr_{init}$ | 1e-3 |
| $wd$ | 1e-5 |
| $bs$ | 64 |
| $drop$ | 0.2 |
| $k$ | 20 |
| $d_{GPE}$ | 128 |
| $d_{hid}$ | 128 |

