# OpenReview forum: "GPEN: Global Positional Encoding Network for Graphs"
_TMLR — Rejected by TMLR_

### Review · Reviewer_txiG · 2024-04-09

**Summary Of Contributions:**

This paper proposes the Global Positional Encoding Network, a new positional embedding method for graph neural networks. The proposed method randomly selects a subset of nodes to be referenced and creates embeddings for each node by random walk. Contrast learning loss is added to the supervised loss so that node embeddings of the same node created by different reference node sets are close and embeddings for different nodes are distinct. The proposed method is applied to node classification tasks using network datasets with various homophily and compared to local distance encoding and other positional encoding methods.

**Audience:**

Yes

**Broader Impact Concerns:**

No broader impact concerns

**Claims And Evidence:**

No

**Requested Changes:**

Q1: In Section 2, PEG is considered a distance-encoding method because it gives weight to edges. Do the authors consider it as a local-distance-aware method as well? If so, it may not be appropriate because it considers the underlying graph's global information by using the Laplacian eigenmaps. For example, two subgraphs with the same topology can have different distance structures depending on how they are embedded in the ambient graph, which results in different representations.

-----------------

Q2: The authors claim that P-GNN is limited in its capability as a general solution compared with GPEG. However, if I do not miss any information, no numerical experiments are provided to support that GPEG solves this problem.

-------------------

Q3, P5: *Given two subgraphs $\mathcal{G}\_1 = (\mathbb{V}\_1, \mathbb{E}\_1)$ and $\mathcal{G}\_2 = (\mathbb{V}\_2, \mathbb{E}\_2)$ separately around the target node $v\_1\in \mathbb{V}\_1$ and $v\_2\in \mathbb{V}\_2$ in $\mathcal{G}$, where $v\_1 \not = v\_2$ and $\mathcal{G}\_1 \not = \mathcal{G}\_2$ there is a function $f\colon \mathbb{Y}\_{u, v} \to \mathbb{R}^n$ that extracts the structural in formation around the $(u, v)$ pair.*: This part has room for improvement. First, $v_1$ and $v_2$ are not used to define $f$. Second, the assumption does not support cases where $v$'s or $\mathcal{G}$'s can be identical, that is, the cases where (1) $v_1 = v_2$ and $\mathcal{G}\_1 \not = \mathcal{G}\_2$ or (2) $v\_1 \not = v\_2$ and $\mathcal{G}\_1 = \mathcal{G}\_2$. Third, since we can choose $f$ for each pair $(v_1, v_2)$, $f$ can depend on $v_1$ and $v_2$. That is, if we consider another pair $(v_1', v_2')$, $f$ used for the pair $(v_1', v_2')$ can be different from $f$ used for the pair $(v_1, v_2)$. I am wondering if the authors intended this dependence.

-------------------

Q4, P6, Eq. (9): The proposed method includes the contrastive loss as a training loss for the supervised task, as Eq. (9) shows. However, this method differs from the usual use of contrastive learning, where we train the model before the downstream task in an unsupervised manner. I want to clarify the reason for designing the learning this way instead of using methods known to be generally successful.

-------------------

Q5, P8, Table 2: Table 2 shows the average of 10 trials only. However, looking at the results in Table 4, I wonder if the trial variance could be high. I want to clarify the standard deviation of 10 trials.

-------------------

Q6, P10, Table 6: *Table 6 shows that [...] it can still outperform P-GNN. The results suggest that GPEN can generalize the graph structure knowledge [...].*: I have a question about the interpretation of the results of the experiments in Table 6. The proposed method indeed outperforms P-GNN. However, as the authors say, the accuracy is better when GPEN is trained from scratch on the target dataset. We should instead interpret this result as a negative transfer of the transfer learning, which limits the effectiveness of transfer learning.

-------------------

Q7, P.10, Table 7: *According to the results in Table 7, PEG-GPEN achieves higher accuracy than using GPEN directly in low $\beta$ value graphs.*: Certainly, PEG-GPEN achieves better accuracy than GPEN in the low $\beta$ regime. However, since it is still worse than PEG, I want to clarify the motivation for employing PEG-GPEN as a global position embedding method.


【Minor Comments】

- P2, Figure 1: Central -> Center
- P3: It is difficult for those unfamiliar with this field to know what PEG is. I would suggest that PEG is a method proposed by Wang et al. (2022)
- P4, Figure 3: The notation of $\boldsymbol{z}_v$ and $\boldsymbol{z}_i$ can be confusing. If we only read Figure 3's caption, it looks that $\boldsymbol{z}_i$ is the embedding for the node $i$, which is not true, as defined in Section 3.3. Also, $\boldsymbol{z}_i'$ is not defined in the preface of Section 3. I would suggest making the descriptions self-contained or referring to the definitions in Section 3.3.
- P5: The notation of $\mathbb{Y}\_{uv}$ needs improvement. Since the variable $u$ is not free on the definiton $\\{y_{uv}\mid u \in \mathbb{V}\\}$, $\mathbb{Y}\_{uv}$ should not depend on $u$. The notation such as $\mathbb{Y}_{v}$ would be preferable. In fact, Definition 3.2 uses this notation.
- P6: *To encode the global position [...], where $f_1$ and $f_2$ are two two-layer MLPs with a non-linearity.*: Remove the first *two*.
- P7, Eq. (10): $n_{same}$ and $n_{neighbor}$ depend on node $v$. Since summation $\sum_{v\in V}$ is summing over $v$, the dependence on $v$ should be explicit (e.g., $n_{same}^v$).
- P.11: *This enhances discrimination with a relatively high degree of consistency.*: I want to clarify what degree of consistency means in this context.

**Strengths And Weaknesses:**

Strengths
- The idea of using contrastive learning for positional encoding of GNN is new.

Weaknesses

- The descriptions of the proposed method have room for improvement, especially in Section 3.2 (See Q3 and Minor Comments).
- The paper claims the shortcomings of existing models (local-distance-aware distance encoding, positional encoding, and P-GNN), e.g., in the introduction. However, if I do not miss any information, the evidence for the claim that proposed method solves these shortcomings is weak, questioning the criterion for TMLR acceptance criteria in terms of claim and evidence (See Q1 and Q2).
- The proposed method does not perform better for heterophilic graphs. Considering that recent GNN models are effective for both homophilic and heterophilic tasks, their applicability is limited.
- I have several questions about the implications of numerical experiments in Tables 6 and 7. (See Q6 and Q7)

---

> ### Author Response · Authors · 2024-05-22
> **Response to Reviewer txiG**
>
> Thank you for the detailed questions. The following are our responses:
>
> This work addresses the shortcomings of encoding methods. It is possible to apply this work to the recent advanced GNNs. Since the reviewer did not identify which GNNs had in mind, we are unable to insert our GPEN into those most recent GNNs for a fair comparison.
>
> A1: The reviewer says PEG is a PE method when using Laplacian eigenmaps. However, we consider PEG a DE method since it falls within the commonly used criteria for methods that use the distances between any node in a graph. In this setting, Fig. 1 (iii) in the PEG paper illustrates that the edges at the symmetric places can have the same weights even if the LE carries global information. Thus, the nodes at symmetric places cannot be distinguished without directly using original node features and Laplacian vectors. The PEG paper only conducts experiments on link prediction tasks, where they "use a GCN layer with edge weights according to the distance between the end nodes of the edge and keep the positional features unchanged.". According to their code https://github.com/Graph-COM/PEG/blob/main/model.py (line 31), they use the squared l2 distance of the positional features as inputs of the last FC layer, along with the node features. When we only use the edge weights generated from the distances of two Laplacian vectors, the network still meets their PE-stable layer conditions. Although it is possible to combine DE and PE methods to achieve better results, it is beyond our research scope since we mainly focus on proposing a PE method. Thus, we classify the PEG as a DE-type method for a simplified and clear comparison in our experiments.
>
> A2: We categorise a method as a general solution if it is parameter-free (i.e. DE-GNN) or can be used as a pre-trained model (i.e. GPEN). According to Eq. 1 in the P-GNN paper, the length of their positional embedding is $\log n \cdot \log n$, which is decided by the total number of nodes in the graph. Thus, P-GNN has to be trained per graph,  which limits its capability as a general solution. On the contrary, the length of positional embedding generated by our GPEN is a hyperparameter independent of the number of nodes. Sec. 4.4.3 shows that our pre-trained GPEN can be directly transferred to different graphs, which makes it a general solution.
>
> A3: We want to express that the target node $v_1$ and $v_2$ are the central node of the subgraph $G_1$ and $G_2$ within $k$ hops. $\{u_{G_1}\}$ and $\{u_{G_2}\}$ are their $k$-hop neighbourhood sets, respectively.In this scenario, the concerns raised by the viewer will not be relevant. We will revise Sec. 3.2 to make presentation cleaner.
>
> A4: Although it is not explicitly the same, our work follows other works using contrastive learning in the supervised tasks: 1. Khosla et al., Supervised Contrastive Learning, NeurIPS, 2020. 2. Wang et al., Exploring Cross-Image Pixel Contrast for Semantic Segmentation, ICCV, 2021. We will add the citations in the revised version. We use contrastive learning to allow the GPEN to capture more robust global positional embeddings. When using cross-entropy loss for classification in the PE setting, we aim to bring nodes with the same labels closer together in the global positional embedding space.
>
> A5: We will include std. in Table 2 and Table 3 in the revised version.
>
> A6: We provide a method that can be used for transfer learning if other people want to. The lower performance is because we only pretrain GPEN with one graph, which causes overfitting and hurts performance in transfer learning. It would be better to achieve generalization if pretraining GPEN on a large-scale dataset with various graphs, which can be investigated in future works.
>
> A7: As we say in the A1, we categorize PEG as a DE method. So, the PEG-GPEN is also a DE method. Our main purpose is to demonstrate GPEN as a PE method that performs better than state-of-the-art on homophilic graphs. In the paper, we point out that the GPEN has limitations on heterophilic graphs. Converting GPEN to a DE method is to show the performance gap between PEG can be shrieked on heterophilic graphs.

---

> > ### Comment · Reviewer_txiG · 2024-05-25
> >
> > I thank the authors for responding to my questions in the initial comments. However, some remain.
> > The following are question-by-question responses:
> >
> > --------------------------
> >
> > Q1
> >
> > > The reviewer says PEG is a PE method when using Laplacian eigenmaps. However, we consider PEG a DE method since it falls within the commonly used criteria for methods that use the distances between any node in a graph.
> >
> > I would argue that I did not claim that PEG is a PE method or deny that it is a DE method. I intended to question whether PEG is a *local* DE method in the sense that the node embedding is determined by the topology of the node's neighbor.
> >
> > > In this setting, Fig. 1 (iii) in the PEG paper illustrates that the edges at the symmetric places can have the same weights even if the LE carries global information.
> >
> > Let us consider the following graph identical to Fig. 1 (iii):
> >
> > ```
> > a     d
> >  \   /
> >   b-e
> >  /   \
> > c     f
> > ```
> >
> > Note that PEG gives the same embeddings to $a$ and $d$ (also, the pairs $(b, e)$ and $(c, e)$). Subgraphs $\{a, b, c\}$ and $\{b, d, e\}$ are isomorphic (i.e., line graphs of size 3). However, the corresponding nodes have different embeddings. It is because the relative position of these subgraphs in the graph, i.e., the global information about how they are embedded in the graph, is used to compute the embeddings. Therefore, I do not think PEG is a local DE method.
> >
> > --------------------------
> >
> > Q2: I thank the authors for explaining the term *general solution*. I understand its usage. I suggest explicitly writing the term's meaning because it is not standard usage. Also, I recommend adding the implication from the definition (cited below) that a model learned on one graph can be transferred to other graphs.
> >
> > > We categorise a method as a general solution if it is parameter-free (i.e. DE-GNN) or can be used as a pre-trained model (i.e. GPEN).
> >
> > --------------------------
> >
> > Q3
> >
> > > We want to express that the target node $v_1$ and $v_2$ are the central node of the subgraph $G_1$ and $G_2$ within $k$ hops.
> >
> > I understand what the authors have in mind. I suggest writing this assumption explicitly.
> >
> > > In this scenario, the concerns raised by the viewer will not be relevant.
> >
> > I respectfully disagree with this claim. Suppose $k$ is greater than the graph's diameter (the distance between the farthest distant node pairs in the graph). Then, no matter which node in the graph is chosen, its $k$-hop neighbor is the entire graph. Therefore, $G_1$ and $G_2$ can be identical even if $v_1$ and $v_2$ are different. Also, the second and third questions still need to be resolved.
> >
> > --------------------------
> >
> > Q4: OK. I understand that it is common to optimize the contrastive loss simultaneously with the supervised loss.
> >
> > --------------------------
> >
> > Q5: OK. I would appreciate it if we could see the standard deviations in the discussion period to see if the variance is sufficiently small to claim significance.
> >
> > --------------------------
> >
> > Q6: I understand what the authors intended. However, I still question whether their claim is appropriate. GPEN is theoretically capable of transfer learning. However, if transfer learning does not reduce prediction performance practically, we cannot say it is generalizable; there is no point in doing it at an additional training cost.
> >
> > --------------------------
> >
> > Q7: I want to clarify what the authors want to claim from this experiment. Since the proposed method is PE, I first thought that the authors wanted to argue that PE methods perform well in heterophilic graphs. However, looking at the following response, this is not true:
> >
> > > Converting GPEN to a DE method is to show the performance gap between PEG can be shrieked on heterophilic graphs.
> >
> > Also, since PEG-GPEN is a DE method according to the authors' criteria, I still question the importance of integrating GPEN with PEG instead of pure PEG, the DE method with higher performance and simpler architecture. Also, from the practitioner's point of view, the importance of the distinction between PE and DE is not clear. We should simply choose the best model regardless of the encoding type.

---

> > > ### Author Response · Authors · 2024-05-29
> > > **Response to Reviewer txiG**
> > >
> > > A1. We want to clarify that LEs are not directly assigned to the input node features during the aggregation in the PEG layer, according to Eq. 6 in [1]. Thus, when we design the PEG baseline, we only use the distance of two LEs as the edge weight $\xi$. The detailed architecture of the PEG baseline we used is presented in Appendix A.2.3 in our paper. In this setting, a GNN will not use the global information inherent in LEs. We also add NAGphormer [2] as another backbone in our revision, directly assigning LEs to the input node features during the aggregation. Based on our results, NAGphormer+GPEN exceeds NAGphormer by a large margin. We are waiting for all the experiments to be finished and will pose here.
> > >
> > > A2. We appreciate the suggestion and will adapt it in the revision for better understanding.
> > >
> > > A3. We think the concern that $k$ is greater than the graph's diameter can be addressed using comparisons on computational graphs instead of input subgraphs. We will make corresponding changes.
> > >
> > > A7. We observe that GPEN can perform better or the same as other encoding methods on heterophilous graphs [3]. So, the defective datasets misled our previous claim about the limitation on heterophilous graphs. As we said in our response to all reviewers, we will replace them.
> > >
> > > [1] Wang et al., Equivariant and Stable Positional Encoding for More Powerful Graph Neural Networks, ICLR, 2022.
> > >
> > > [2] Chen et al., NAGphormer: A Tokenized Graph Transformer for Node Classification in Large Graphs, ICLR, 2023.
> > >
> > > [3] Platonov et al., A critical look at evaluation of GNNs under heterophily: Are we really making progress?, ICLR, 2023.

---

> > > > ### Comment · Reviewer_txiG · 2024-05-30
> > > >
> > > > I thank the authors for the response.
> > > >
> > > > Q1
> > > >
> > > > > We want to clarify that LEs are not directly assigned to the input node [...]. Thus, when we design the PEG baseline, we only use the distance of two LEs as the edge weight $\xi$. [...] A GNN will not use the global information inherent in LEs in this setting.
> > > >
> > > > I am afraid I respectfully disagree with this claim. My understanding is that LE uses global information. Therefore, the PEG edge weight $\xi$ calculated from it also uses (depends on) global information in general. If we want to claim that $\xi$ is free from global information, it requires some evidence (e.g., Formulate the claim mathematically and prove it. Verify it by numerical experiments).
> > > >
> > > > > We also add NAGphormer [2] as another backbone in our revision, directly assigning LEs to the input node features during the aggregation.
> > > >
> > > > I understand. I am looking forward to the experiment results.
> > > >
> > > > ---------------------------
> > > >
> > > > Q2: OK
> > > >
> > > > ---------------------------
> > > >
> > > > Q3
> > > >
> > > > I will wait for and check the revision to see if my concerns are solved.
> > > >
> > > > ---------------------------
> > > >
> > > > Q7
> > > >
> > > > > We observe that GPEN can perform better or the same as other encoding methods on heterophilous graphs [3].
> > > >
> > > > I guess this claim is based on the results of Wikipedia's squirrel, chameleon, and actor datasets (point out if I am wrong). However, [3] (Platonov et al., 2023) also points out problems with datasets in Wikipedia and WebKB. Therefore, I am wondering why the authors replaced WebKB but not Wikipedia.
> > > >
> > > >
> > > > ---------------------------
> > > >
> > > > I am also wondering what the authors think about Q6.
> > > >
> > > > Best,
> > > > Reviewer txiG

---

> > > ### Author Response · Authors · 2024-06-10
> > > **Response to txiG**
> > >
> > > A1. We want to add that Eq.4 in our paper is for introducing the selecting referential node set to generate inputs for GPEN. This part is not intended to explain the limitations of DE-GNN or PEG.

---

> > > > ### Comment · Reviewer_txiG · 2024-06-13
> > > >
> > > > I thank the authors for the additional comments and clarification. I understand the authors' intention.

---

> ### Author Response · Authors · 2024-05-31
> **Response to Reviewer txiG**
>
> A1: the following is an example where L2 distances of LEs lose global information.
>
> ```
> a [0.39,-0.30]                    d [-0.39, 0.64]
>  \ 0.09                          / 0.41
>    \              0.8          /
>    b [0.45, 0.00] -----e [-0.45, 0.00]
>   / 0.09                      \ 0.41
> c [0.39, 0.30]                  f [-0.39, -0.64]
> ```
> [,] denotes the LE. The values on the edges denote the L2 distances of LEs. As the example shows $\xi_{ab} = \xi_{bc}$ and $\xi_{de} = \xi_{ef}$.
>
> A7: Yes, the Wikipedia will also be removed. Sorry for not mentioning it in the previous response.

---

> > ### Comment · Reviewer_txiG · 2024-05-31
> >
> > Dear authors
> >
> > Thank you for the response
> >
> > A1.
> >
> > Thank you for your detailed explanation. The authors' example is exactly what I think is the counterexample, as I wrote in the [previous response](https://openreview.net/forum?id=XSfU9bmZnN&noteId=1NI80B1I1G)
> >
> > > Subgraphs ${a, b, c}$ and ${b, d, e}$ are isomorphic (i.e., line graphs of size 3). However, the corresponding nodes have different embeddings.
> >
> > I guess the authors and I seem to use the term *global information* with different meanings. In my understanding, the relative position of a graph in the whole position can be considered as global information. Edge weights use this information through LE, and the subgraphs ${a, b, c}$ and ${b, d, e}$ have different edge weights.
> >
> > ---------------------
> >
> > A7: OK

---

> > > ### Author Response · Authors · 2024-05-31
> > > **Response to Reviewer txiG**
> > >
> > > Thank you for the clarification.
> > >
> > > What we want to say is that distance encoding such as $\xi$ in PEG will provide less information during the aggregation when distinguishing a from c, or d from f.

---

### Review · Reviewer_87ko · 2024-04-14

**Summary Of Contributions:**

This paper proposes a global positional encoding method for  graph neural networks (GNNs). It chooses a subset of nodes as referential nodes and then, for any node $v$, it use the random walk probabilities to these referential nodes as a vector of positional encoding for $v$.

**Audience:**

Yes

**Broader Impact Concerns:**

Unclear if the proposed method is of  any use in SOTA GNN models.

Otherwise,
n/a

**Claims And Evidence:**

No

**Requested Changes:**

1. More extensive evaluation using SOTA GNN models as a baselines

2. Illustration of  the usefulness of the proposed node encoding method
for  downstream graph learning tasks such as node classification, graph classification, and/or link prediction.

2. Improving presentation.

**Strengths And Weaknesses:**

Strengths:

-  The idea is simple, which can be seen as a generalization of the distance encoding of DE-GNN (2020).
The paper is somewhat interesting.

Weaknesses:

1. The evaluation is superficial. The node embedding or encoding is mainly used for learning tasks such as  node classification, graph classification, link prediction, and graph structural learning. This paper in its current form only compares with 3-4 very basic methods or models. It does not consider comparing with SOTA baselines, e.g., those methods based on
transformer structures. A simple google search may return various different SOTA methods.

Without extensive comparisons, it is unclear if this node encoding method would indeed play an important role in SOTA models. In particular, some transformer-based models use positional encoding already. Whether this proposed method is useful or  not needes to be tested.

2. The presentation is problematic. There are still quite some notations to be clarified and editorial errors to  be corrected. For example:

- Page 5, line 4 of Section 3.2, regarding the definition of $f$:
It maps to $R^n$. What is it? No explanation at all.

- Page 5, first line after eq. (4), regarding definition of $Y_{uv}$:
Why does  this definition only has $u \in V$? What is $v$?

---

> ### Comment · Reviewer_87ko · 2024-04-14
> **Unclear if the proposed method would be useful**
>
> This paper proposes a global positional encoding method for  graph neural networks (GNNs).
> It chooses a subset of nodes as referential nodes and then, for any node $v$, it use the
> random walk probabilities to these referential nodes as a vector of positional encoding for $v$.
>
> The idea is simple, which can be seen as a generalization of the distance encoding of DE-GNN (2020).
> The paper is somewhat intereting. However, there are some major concerns:
>
> 1. The evaluation is superficial. The node embedding or encoding is mainly used for learning tasks
> such as  node classification, graph classification, link prediction, and graph structural learning.
> This paper in its current form only compares with 3-4 very basic methods or models. It
> does not consider comparing with SOTA baselines, e.g., those methods based on
> transformer structures. A simple google search may return various different SOTA methods.
>
> Without extensive comparisons, it is unclear if this node encoding method would indeed
> play an important role in SOTA models. In particular, some transformer-based models use
> positional encoding already. Whether this proposed method is useful or  not needes to be tested.
>
> 2. The presentation is problematic. There are still quite some notations to be clarified and
> editorial errors to  be corrected. For example:
>
> - Page 5, line 4 of Section 3.2, regarding the definition of $f$:
> It maps to $R^n$. What is it? No explanation at all.
>
> - Page 5, first line after eq. (4), regarding definition of $Y_{uv}$:
> Why does  this definition only has $u \in V$? What is $v$?
>
> Based on the review results, I would recommend review after major revision, or rejection in its current form.

---

> ### Author Response · Authors · 2024-05-22
> **Response to Reviewer 87ko**
>
> About SOTA models:
>
> We are adding a transformer-based method as a backbone [1].
>
> Presentation issues:
>
> 1. In Page 5, line 4 of Section 3.2, $f$ is the mapping function a GNN learnt that extracts the structural information around the $(u, v)$ pair.
>
> 2. $v$ is either the target node $v_1$ or $v_2$.
>
> We will revise Sec. 3.2 to make presentation cleaner.
>
> [1] Chen et al., NAGphormer: A Tokenized Graph Transformer for Node Classification in Large Graphs, ICLR, 2023.

---

> > ### Comment · Reviewer_87ko · 2024-06-21
> > **want to see more experimental results**
> >
> > I would like to thank the authors for the rebuttal to my comments and questions.
> > After reading the rebuttal and the revised paper, I see the authors added one more model based on Transformers for comparison. However, I still feel that the evaluation is not so satisfactory. I think that comparison with more models, in particular, some baseline models based on Transformers and with explicit position embedding will help readers understand the advantages of the proposed method.

---

### Review · Reviewer_827F · 2024-05-14

**Summary Of Contributions:**

The paper proposes Global Positional Encoding Network (GPEN), which extends the range of structural information from local subgraphs to the entire graph by calculating the distances from each node to a set of randomly sampled referential nodes. Contrastive loss on pairwise distances of different nodes is employed to make positional representations more discriminative while retaining the relative interactions between nodes. GPEN is adaptable to any GNN model for diverse tasks. The experimental results demonstrate that by inserting the encoding scheme into a backbone GNN, the proposed method can outperform state-of-the-art encoding methods. In particular, it surpasses other distance encoding and positional encoding-based models on homophilic graph datasets.

**Audience:**

Yes

**Claims And Evidence:**

No

**Requested Changes:**

1. (Referring to weakness #1) More illustrations are expected on why randomly selected referential nodes is good enough for GPEN to serve as a coordinate axis that breaks the similarity of the local structure.
2. (Referring to weakness #2) More explanations are expected on why GPEN fails to generate distinguishable embeddings on Actor and Cornell.
3. (Referring to weakness #3) The calculation time or the time complexity of calculating the distances could be provided to verify the efficiency on large datasets.
4. (Referring to weakness #4) It would be better to test the proposed method on different GNN backbones and to compare with more recent distance-encoding methods.

**Strengths And Weaknesses:**

Strengths:

1. The proposed method enables global structural information, and avoids ambiguity when comparing two subgraphs with the same structure but located in different parts of the same graph.
2. The proposed global position encoding is adaptable to various backbone architectures.
3. The experimental results demonstrate that the proposed GPEN obtains either higher or comparable accuracy on most datasets, which might suggest the effectiveness and efficiency of the proposed model.

Weaknesses:
1. As stated in Section 3.2, the model is based on the fact that when the referential nodes are properly selected, they can serve as a coordinate axis that breaks the similarity of the local structure. However, the proposed GPEN only adopts randomly selected nodes as referential nodes. Although the contrastive loss aims to enable GPEN to learn unique global positional embeddings for all nodes in a graph, whether the selection of referential nodes is proper and is able to provide enough distance information to represent the whole graph is yet to be examined.
2. As stated in Section 4.4.1, GPEN fails to provide distinguishable positional embeddings on Actor and Cornell, as nodes from different classes are evenly distributed in the embedding space. These two datasets are with a rather large (Actor) or small (Cornell) number of nodes. Does this observation suggest the proposed GPEN might fail to provide enough structural information to distinguish different nodes on extremely large or small datasets?
3. The computational cost of calculating the distances between all nodes in a graph and the selected referential nodes might be rather high on large graphs.
4. The chosen baselines are somewhat outdated. For instance, DE-GNN was published in 2020, and P-GNN in 2019. It would enhance the convincingness if comparing the proposed model with more recent distance-encoding works, e.g., PathNet [1].

[1] Sun Y, Deng H, Yang Y, Wang C, Xu J, Huang R, Cao L, Wang Y, Chen L. Beyond Homophily: Structure-aware Path Aggregation Graph Neural Network. In IJCAI 2022 (pp. 2233-2240).

---

> ### Author Response · Authors · 2024-05-22
> **Response to Reviewer 827F**
>
> About the referential nodes sampling strategy:
>
> Our current method saves the selection for each graph when the model achieves the best results on the validation set. To evaluate the robustness of the trained model under different referential node selections, we show the standard deviations of the accuracy when the referential nodes change, which indicates that the referential node changes only have a subtle impact in most cases.
>
> About the computational cost:
>
> We will add computational cost comparison with all PE and DE methods.

---

> > ### Comment · Reviewer_827F · 2024-06-10
> > **Thank you for the response**
> >
> > I decide to maintain my recommendation. The additional experimental results do not address my concerns on the performance on specific datasets and the selection of referential nodes. The additional experiments haven't included the comparison between the proposed model and more recent distance-encoding works, e.g., PathNet.

---

### Author Response · Authors · 2024-05-22
**To all reviewers**

About experiments:

Because WebKB has issues such as extremely unbalanced labels and too small graphs, we are replacing it with the heterophilous datasets proposed by [1] and adding a transformer-based method proposed by [2]. The experiments are in progress. We will post the results of the main experiments here and make changes in the revised version.

About notations and definitions:

We will address issues and make them cleaner in the revised version.

[1] Platonov et al., A critical look at evaluation of GNNs under heterophily: Are we really making progress?, ICLR, 2023.

[2] Chen et al., NAGphormer: A Tokenized Graph Transformer for Node Classification in Large Graphs, ICLR, 2023.

---

> ### Author Response · Authors · 2024-06-07
> **To all reviewers**
>
> The following are the new results of our main experiments. We are modifying our paper according to the latest results and reviewers' responses.
>
> Accuracy (%) is reported for Cora, Citeseer, Roman-empire and Amazon-ratings. ROC AUC (%)
> is reported for Minesweeper and Tolokers. The number in the bracket indicates the ranking of the performance.
>
> | Type | Dataset                | Cora             |                  | Cite.            |                  | Roma.            |                  | Amazon.          |                  | Mine.            | Tolokers         |                  |
> |------|------------------------|------------------|------------------|------------------|------------------|------------------|------------------|------------------|------------------|------------------|------------------|------------------|
> |      | Adjusted homophily     | 0.77             |                  | 0.68             |                  | -0.05            |                  | 0.14             |                  | 0.01             | 0.09             |                  |
> |      | Label informativeness  | 0.59             |                  | 0.46             |                  | 0.11             |                  | 0.04             |                  | 0.00             | 0.01             |                  |
> |      | Attr.                  | w/               | w/o              | w/               | w/o              | w/               | w/o              | w/               | w/o              | w/               | w/               | w/o              |
> | None | GraphSAGE              | 86.12 ± 0.96 (3) | 41.41 ± 2.54     | 76.12 ± 1.88     | 33.26 ± 1.73     | 81.73 ± 0.50     | 30.67 ± 0.79     | 49.76 ± 0.41 (3) | 36.97 ± 0.31     | 90.82 ± 0.62     | 83.54 ± 0.70     | 69.92 ± 1.58     |
> | DE   | GraphSAGE-DE-GNN       | 86.30 ± 1.72 (2) | 41.01 ± 1.29     | 76.45 ± 1.92 (2) | 24.25 ± 2.38     | 82.12 ± 0.43 (3) | 36.42 ± 0.60 (1) | 50.06 ± 0.77 (1) | 37.65 ± 0.54     | 90.93 ± 0.79 (1) | 83.48 ± 0.84     | 68.82 ± 1.00     |
> | DE   | GraphSAGE-PEG          | 85.09 ± 1.72     | 41.45 ± 2.27     | 76.58 ± 1.79 (1) | 31.14 ± 2.77     | 81.66 ± 0.64     | 31.84 ± 0.49     | 49.45 ± 0.65     | 36.99 ± 0.24     | 90.86 ± 0.64 (2) | 83.26 ± 0.70     | 68.75 ± 1.21     |
> | PE   | GraphSAGE-P-GNN        | 85.33 ± 1.91     | 59.78 ± 2.38 (3) | 75.03 ± 2.32     | 46.68 ± 5.26     | 82.28 ± 0.28 (2) | 32.05 ± 0.28     | 50.05 ± 0.49 (2) | 41.88 ± 0.69 (2) | 90.59 ± 0.20     | 83.72 ± 0.35 (3) | 75.53 ± 0.58 (3) |
> | PE   | GraphSAGE-GPEN (Ours)  | 86.34 ± 1.36 (1) | 75.27 ± 2.05 (2) | 76.19 ± 1.56 (3) | 51.49 ± 5.54 (1) | 83.79 ± 0.69 (1) | 36.39 ± 0.44 (2) | 49.42 ± 0.32     | 38.41 ± 0.99 (3) | 90.83 ± 0.73 (3) | 84.42 ± 0.79 (1) | 79.77 ± 0.96 (1) |
> | PE   | NAGphormer             | 83.72 ± 2.04     | 52.60 ± 2.14     | 74.15 ± 2.27     | 50.07 ± 5.37 (3) | 77.02 ± 0.78     | 21.08 ± 0.47     | 44.10 ± 0.66     | 42.26 ± 0.52 (1) | 86.22 ± 1.40     | 82.52 ± 1.04     | 74.39 ± 0.90     |
> | PE   | NAGphormer-P-GNN       | 83.78 ± 1.66     | 39.48 ± 6.16     | 74.10 ± 2.42     | 44.23 ± 5.34     | 77.37 ± 0.7      | 22.13 ± 1.11     | 43.63 ± 0.84     | 38.11 ± 1.99     | 87.40 ± 1.46     | 82.86 ± 1.16     | 74.22 ± 2.07     |
> | PE   | NAGphormer-GPEN (Ours) | 84.08 ± 1.86     | 77.20 ± 2.09 (1) | 74.33 ± 1.44     | 51.15 ± 5.95 (2) | 80.12 ± 0.48     | 34.56 ± 0.50 (3) | 43.25 ± 0.60     | 37.87 ± 1.25     | 86.79 ± 1.16     | 84.38 ± 0.92 (2) | 79.68 ± 1.14 (2) |

---

> ### Author Response · Authors · 2024-06-10
> **To all reviewer**
>
> We have uploaded a new version based on the response about Sec. 3 and new experiments. Because we did not observe meaningful visualization results on the heterophilic datasets, we have removed the visualization. Additionally, GPEN already achieved reasonable results on the new datasets. Thus, we also removed PEG+GPEN experiments because we did not observe improvements.
>
> We will soon add details of NAGphormer to the appendix.

---

> > ### Author Response · Authors · 2024-06-11
> > **To all reviewer**
> >
> > We have uploaded the final version.

---

### Decision · Action_Editor_6w3k · 2024-06-22

**Recommendation:** Reject

**Comment:**

Dear Authors,

Thank you for submitting your manuscript. After careful consideration of the reports from three reviewers, we regret to inform you that we have decided to reject your submission.

While your efforts to address the reviewers' initial comments are appreciated, there are still significant concerns that remain unresolved. The evaluation is deemed insufficient, particularly lacking a broader comparison with baseline models based on Transformers with explicit position embeddings. Additionally, despite improvements in the mathematical descriptions, inappropriate formulations persist, and the numerical experiments do not convincingly support the paper's claims. Specifically, the ablation studies presented are not strong enough to substantiate the stated benefits of InfoNCE loss and transfer learning for GPEN. Furthermore, concerns about the performance on large and small datasets were not adequately addressed, and the rationale for adopting randomly selected referential nodes remains unclear. More related baselines are needed for a comprehensive comparison.

We encourage you to consider these points carefully and revise your work accordingly. We hope the feedback provided will be helpful in improving your research.

With kind regards,

Moshe.

**Audience:**

Most reviewers agree that the audience of TMLR would fine interest in the finding of this paper, and I agree with them.

**Claims And Evidence:**

The reviewers are not fully convinced of the evidence of claims in the submission. Despite many clarifications made in the revision stage, there are still points that the reviewers are not satisfied with. In particular:

1. A comparison with more models is required.  In particular, some baseline models based on Transformers and with explicit position embedding will help readers understand the advantages of the proposed method.

2. Some of the formulations are still inappropriate.

3. The ablation study in Section 4.4.1 is not strong enough to claim that the InfoNCE loss enhances GPEN's stability. In addition,  the ablation study in Section 4.4.2 does not seem to show the effectiveness of the transfer learning of GPEN.

4. The authors' response has not adequately addressed the concern on the performance on large and small datasets. The authors illustrate the reason of adopting randomly selected referential nodes only from the experimental results, which does not clarify the concern on how the referential nodes can serve as a coordinate axis that breaks the similarity of the local structure. The new imported baseline, NAGphormer, is not specific for distance-encoding feature. The authors should include more related baselines for comparison.

**Resubmission Of Major Revision:**

The authors may consider submitting a major revision at a later time.